# Structural basis of transcriptional regulation by a nascent RNA element, HK022 *put*RNA

**Seungha Hwang** [1], **Paul Dominic B. Olinares** [2], **Jimin Lee**[1], **Jinwoo Kim** [1], **Brian T. Chait** [2], **Rodney A. King**[3] & **Jin Young Kang** [1]✉

Transcription, in which RNA polymerases (RNAPs) produce RNA from DNA, is the first step of gene expression. As such, it is highly regulated either by *trans*-elements like protein factors and/or by *cis*-elements like specific sequences on the DNA. Lambdoid phage HK022 contains a *cis*-element, *put*, which suppresses pausing and termination during transcription of the early phage genes. The *put*RNA transcript solely performs the anti-pausing/termination activities by interacting directly with the *E.coli* RNAP elongation complex (EC) by an unknown structural mechanism. In this study, we reconstituted *put*RNA-associated ECs and determined the structures using cryo-electron microscopy. The determined structures of *put*RNA-associated EC, *put*RNA-absent EC, and σ70-bound EC suggest that the *put*RNA interaction with the EC counteracts swiveling, a conformational change previously identified to promote pausing and σ70 might modulate *put*RNA folding via σ70-dependent pausing during elongation.

Most viruses utilize the host transcription apparatus to express their genes, and viral genomes contain assorted *cis*- and *trans*-elements that manipulate the host transcription machineries[1,2]. Most early genes of lambdoid phages are preceded by transcription terminators; therefore, the host transcription apparatus must be converted to a terminator-resistant form to promote full gene expression of viral genes[1,3,4]. For λ, anti-termination is promoted by the virus-encoded N protein, which binds to the *cis*-acting *nut* sites and suppresses transcription termination[5–7]. By contrast, bacteriophage HK022, discovered in Hongkong in the early 1970s, is related to λ phage but only requires *cis*-acting RNAs, named *put* (polymerase-utilization), to promote read through of transcription terminators without any dedicated *trans*-acting protein factors[8,9]. *Put*-mediated anti-termination is efficient and robust, and has been shown to suppress both intrinsic and ρ-dependent transcription terminations[9–11].

The HK022 genome contains two *put*RNAs—*put*L and *put*R, located downstream of the early promoters $P_L$ and $P_R$, respectively[11]. Both *put*RNAs are ~70-nt long, share sequence similarity, and are composed of two stem-loop structures[12]. Whereas *put*L is located just downstream of the $P_L$ promoter, *put*R is located ~270-nt downstream of the

$P_R$ transcription start site[11]. The anti-termination activity of the *put*RNAs persists even on terminators located at least 10 kb away and is not dependent on the tethering between the *put*RNA and the elongation complex (EC) via the nascent transcript, suggesting that *put*RNA itself can remain stably associated with the EC as elongation proceeds[13]. The activity of *put*RNA is blocked by the host RNA polymerase (RNAP) mutations that are located exclusively in β′ zinc-binding domain (β′ZBD)[10,14,15]. This genetic evidence together with biochemical evidence suggests that *put*RNA interacts with the RNAP via the β′ZBD[16].

In addition to anti-termination activity, *put*RNA exhibits anti-pausing activity. *Put*L inhibits backtracking at a pause located 21-nt downstream of the second stem (stem II) of *put*L, and this activity is abrogated by the insertion or deletion of several bases between the *put*L and the pausing site[17]. The dependency on the distance between *put*RNA and its location of action suggests that the anti-pausing and anti-termination activities of *put*RNA differ mechanistically. Interestingly, *put*RNA reduces both backtrack and hairpin-dependent pauses like RfaH[18]. RfaH, a paralog of NusG, recognizes an *ops* (operon polarity suppressor) sequence on the non-template DNA strand loaded onto

[1]Department of Chemistry, Korea Advanced Institute of Science and Technology (KAIST), Daejeon, Republic of Korea. [2]Laboratory of Mass Spectrometry and Gaseous Ion Chemistry, The Rockefeller University, 1230 York Avenue, New York, NY, USA. [3]Biology Department, Western Kentucky University, Bowling Green, KY, USA. ✉e-mail: jykang59@kaist.ac.kr

the RNAP EC, changes its C-terminal helices into a β-sheet KOW domain fold to become active, and inhibits transcriptional pausing by resisting RNAP swiveling[19,20].

A structural analysis of the *put*RNA-associated EC is required to understand the molecular mechanism of anti-pausing and anti-termination activities of the *put*RNA. In this study, we synthesized the *put*L RNA using the *Eco* RNAP σ[70]-holoenzyme initiated from the native HK022 P_L promoter, and captured the modified ECs using a transcription roadblock for cryo-EM analysis. We observed *put*RNA-associated EC (*put*EC), *put*RNA-absent EC (*put*-less EC), and σ[70]-bound EC that contains intact *put*RNA (σ[70]-bound *put*EC). Comparison between *put*EC and *put*-less EC structures revealed that the *put*RNA binding to the β'ZBD hinders pausing by reducing the swiveling motion of the EC. Additionally, the σ[70]-bound *put*EC structure suggested that σ[70] binding to EC might facilitate RNA folding as well as play a role in transcription modulation.

## Results

### Preparation and examination of *put*EC

Because an active form of HK022 *put*RNA can be produced only by the enzymatic synthesis using host RNAP, we prepared the *put*RNA-associated EC by initiating RNA synthesis with *Eco* RNAP holoenzyme and stalling the synthesis using a roadblock protein LacI (*lac* repressor), as previously described with some optimization for cryo-EM study (Fig. 1)[16]. Briefly, we first synthesized a DNA scaffold including HK022 P_L promoter, the *put*L sequence, and the *lacO* sequence (LacI binding sequence) (Fig. 1a). Hereafter, we will use '*put*RNA' to denote *put*L RNA for convenience. Holoenzyme containing *Eco* core RNAP and σ[70] was added to the DNA scaffold to form an open complex, and LacI was added to bind to the *lacO* sequence on the DNA as a roadblock (Fig. 1b). Upon rNTP addition, RNAP synthesized the *put*RNA and stalled on the DNA at the roadblock. Excess rNTP was then removed by gel filtration column to prevent further RNA synthesis, and isopropyl β-D-thiogalactopyranoside (IPTG) was added to release the LacI from the DNA scaffold. The complexes were concentrated for cryo-EM grid preparation as well as native mass spectrometry (nMS) analysis.

To stall the EC at a site where the RNAP pauses in the absence of *put*RNA, we generated multiple DNA scaffolds having various distances between the pausing site and the *lacO* sequence, and performed radiolabeled transcription assay with these scaffolds. For screening, we used a *put*⁻ DNA scaffold that allows transcriptional pausing at the pausing site as a control (Fig. 1c, Supplementary Note 1, Supplementary Fig. 1, Supplementary Table 1)[11]. From the screening, we chose a scaffold that has a 7-nt spacer between the pausing site and the *lacO* sequence (7-nt scaffold), and then attempted to analyze the assembled *put*EC by nMS using the same workflow as in our previous nMS studies of bacterial ECs[21–23]. However, we were unable to observe the fully assembled *put*EC due most likely to sample instability or sample heterogeneity and adduction on the long, exposed nucleic acid scaffold during nMS analysis. Nevertheless, nMS analysis of the RNAs extracted from the reconstituted *put*EC revealed two main populations—one RNA was synthesized until the known pausing site (C94), and the other RNA extended by 1-nt (U95) (Fig. 1d)[11,17]. The quantity of the U95 RNA was roughly 1.5 times more than the C94 RNA; therefore, we modeled the U95 RNA placing the U95 nucleotide at the *i+1* site. Since the C94 and U95 nucleotides are located at the *i* and *i+1* sites, the modeled structures would exhibit the same RNA-DNA register both for the C94 and the U95 RNA-containing ECs at the post- and pre-translocated states, respectively (See below).

### Cryo-EM structures of the *put*EC in three conformations

Cryo-EM analysis of the *put*EC prepared by promoter-dependent transcription initiation, transcription elongation, and LacI roadblocking revealed three EC populations at sub-4 Å resolution: (1) the *put*EC (3.2 Å-resolution, 36.9% of the EC particles) that contains

well-folded *put*RNA, (2) the *put*-less EC (3.6 Å, 22.4%) that does not display any well-defined *put*RNA density and (3) the σ[70]-bound *put*EC (3.6 Å, 40.7%) that contains both σ[70] and *put*RNA (Table 1, Supplementary Figs. 2 and 3). We also observed a population consisting of the RNAP holoenzyme loaded onto the template DNA. This population probably resulted from abortive initiation and generated a 3.0 Å-resolution map. This complex is not discussed here because the structures of the holoenzyme open complex have been described in previous reports[24–26].

In the cryo-EM structure of the *put*EC, the *put*RNA was located at the opening of the RNA exit channel of the EC adjacent to the β'ZBD (Fig. 2a). This location is consistent with the *put*-inactivating RNAP mutations and potentially would restrict the RNA hairpin formation in the adjoining RNA exit channel via electrostatic repulsion (Supplementary Fig. 4)[10,27]. The quality of the cryo-EM map allowed us to build the highly-structured *put*RNA de novo (estimated local resolution of the cryo-EM map around the *put*RNA was ~3.5 Å; Fig. 2b, c, Supplementary Note 2, Supplementary Figs. 5–7). The modeled *put*RNA from U2* to U74* contains twenty-one Watson-Crick (WC) base pairs and five non-canonical base pairs, A9*-G35* (Saenger class VIII), G12*-U32* (Saenger class XXVIII), G43*-A64* (Saenger class XI), G42*-U65* and C44*-A63* (Supplementary Fig. 8). To distinguish the nucleic acid residues of the *put*RNA from the amino acid residues in the RNAP, we have added an asterisk (*) to the residue number of the *put*RNA throughout this manuscript. Surprisingly, the cryo-EM structure of the *put*RNA was different from previously published data[11,12] as follows (Fig. 2b–d): First, the 5'-end of the *put*RNA is not C10* but A3*. In the structure, the *put*RNA region from A3* to G7* makes an RNA duplex with the opposite strand from U19* to C15*. Interestingly, this corresponds to the result of the *put*L V1 RNase reaction, which suggested the presence of RNA duplex in the upstream of C10*[12]. Furthermore, this RNA duplex interacts with another RNA strand from U21* to C25* forming an unexpected minor groove RNA triplex structure[28]. The deletion of the third RNA strand, Δ20*−23*, decreased the anti-termination activity of the *put*RNA by ~50%[12], indicating that the triple helix region has a significant effect on the function of the *put*RNA. Second, G35*, which was expected to be located at the bifurcation point of the two stem regions in an unpaired state, base pairs with A9*. This G35*-A9* base pair provides a platform for β'ZBD binding and stabilizes the overall structure of the *put*RNA. This base pair explains why the G35*U mutation retained 70% of the anti-termination activity while the G35*A mutation completely abolished the activity[12]. Meanwhile, A9*C mutation abolished the in vivo anti-termination activity, implying complicated effects of mutations on the G35*-A9* base pair[12]. Third, the *put*RNA contains a bulged loop region (from C26* to G29*) in the middle of stem I, in contrast to the prediction that stem I has a loop region at the end of the stem I ranging from C18* to C26*. This region also provides an interface for binding to the RNAP. At last, the middle region of stem II exhibits distinct base pairings compared to the predicted structure. The middle region of stem II in the cryo-EM structure contains three non-canonical base pairs with three unpaired bases instead of having one non-canonical base pair with five unpaired bases in the predicted structure. This region has relatively high local resolution indicating its structural stability, and makes interfaces with RNAP and the stem I of the *put*RNA.

### Interactions between the *put*RNA and the EC

In the *put*EC structure, the 'V'-shaped *put*RNA binds to the prominent β'ZBD by its pothole formed in the center of the 'V' (Fig. 3a). The β'ZBD fits snugly to the *put*RNA surface, generating a 1130.3 Å² interface area formed by ~33% of the total *put*RNA residues[29]. At the backside of the *put*RNA-β'ZBD interface, the N-terminal loop of the β flap-tip helix makes significant contact with the *put*RNA with an interface area of 281.6 Å². Most of the potential interactions between the *put*RNA and the RNAP comprise polar interactions such as salt bridges, hydrogen

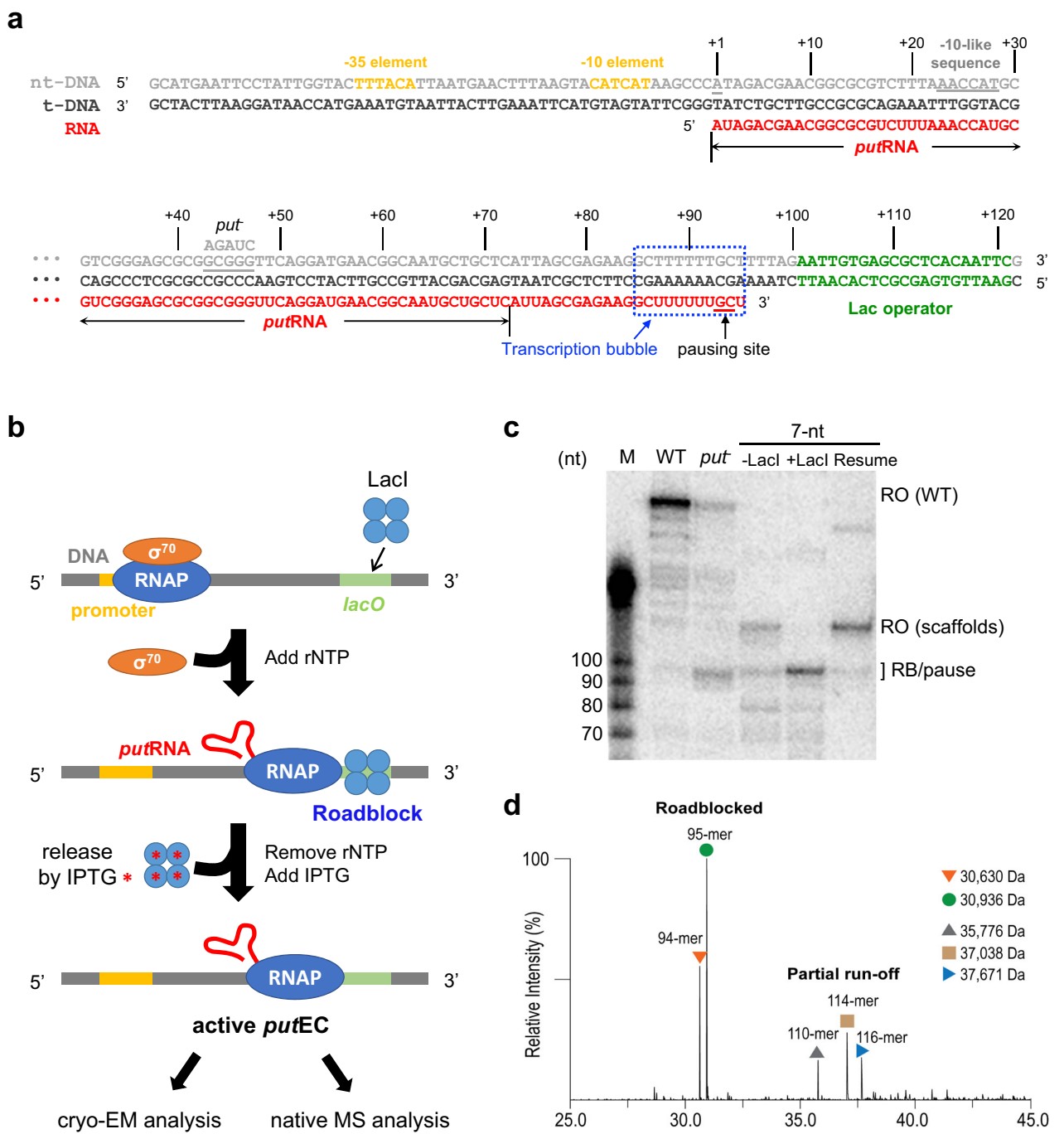

**Fig. 1 | Scaffold design, reconstitution strategy, and evaluation for the *put*EC.**
**a** The nucleic acid scaffold in the reconstituted *put*EC. The DNA was synthesized by PCR, and the RNA is synthesized in vitro from *Eco* RNAP holoenzyme by adding rNTP as described in **b**. The transcription regulation elements including −35 element, −10 element, *put*RNA, −10-like sequence, the pausing site, and *lacO* sequence for roadblocking are shown. Transcription bubble is marked with blue dotted box. Transcript numbering is written for reference. **b** A schematic diagram of the *put*EC reconstitution. First, *Eco* holoenzyme containing *Eco* RNAP and σ70 binds to the promoter region of the DNA forming RPo. LacI is added to the RPo to make a roadblock at the *lacO* site and rNTP is added to initiate the transcription. After transcribing *put*RNA, the RNAP stalls due to the roadblocking LacI. Then, free rNTP is removed by using size-exclusion column before LacI is detached by IPTG to prevent further RNA synthesis. The prepared, active *put*EC is used for cryo-EM and nMS analyses. **c** Radiolabeled transcription assay reveals that the wild-type (WT) *put*

sequence (from pRAK31[11]) did not display paused transcript at the pausing site (lane 2) while the *put*− sequence showed the paused transcript band (lane 3). The designed DNA scaffold, 7-nt, did not show pausing in the absence of LacI, but stalled at the pausing site when LacI was added (Lane 4, 5). For the lanes 2–5, the transcription reaction was done for 2 min at 37 °C. To show the roadblocked EC is still capable of transcription, we added 2 mM IPTG, incubated for 2 min, and resumed the transcription by adding additional rNTP. The resumed reaction mixture was quenched after 2 min and loaded onto the gel (Lane 6, labeled 'Resume'). Details are in the Method section. RO stands for run-off. **d** The RNA extracted from the reconstituted *put*EC is analyzed by nMS. The result revealed two main peaks of C94 and U95, labeled as 94-mer and 95-mer, respectively. The ratio between 94-mer and 95-mer was approximately 2:3. Partial run-off products were observed at higher mass region. Source data are available as a Source Data file.

## Table 1 | Cryo-EM data collections, processing, and validation

| | *put*EC (PDB 7XUE/ EMD-33466) | *put*-less EC (PDB 7XUG/ EMD-33468) | σ⁷⁰-bound *put*EC (PDB 7XUI/ EMD-33470) |
|---|---|---|---|
| **Data collection and processing** | | | |
| Microscope/camera | TFS Krios G4/Gatan K3 BioQuantum | | |
| Voltage (kV) | 300 | | |
| Data acquisition software | EPU version 2.6.0 | | |
| Electron exposure (e⁻/Å²) | 42.16 | | |
| Defocus range (μm) | −2.6 to −0.8 | | |
| Pixel size (Å) | 1.06 | | |
| Symmetry imposed | C1 | | |
| Initial particle images (no.) | 1,102,600 | | |
| Final particle images (no.) | 120,100 | 103,900 | 88,700 |
| Map resolution (Å) | 3.17 | 3.57 | 3.61 |
| FSC threshold | 0.143 | | |
| Map sharpening B factor (Å²) | −63.94 | −78.56 | −67.56 |
| **Model composition** | | | |
| Non-hydrogen atoms | 27864 | 26026 | 29610 |
| Protein residues | 3199 | 3172 | 3473 |
| Nucleotide residues | 136 | 60 | 111 |
| Ligands (Zn²⁺/Mg²⁺) | 2/1 | 2/1 | 2/1 |
| **B factors (Å², mean)** | | | |
| Protein | 66.26 | 68.14 | 101.33 |
| Nucleotide | 128.41 | 140.08 | 207.20 |
| **RMSDs** | | | |
| Bond length (Å) | 0.004 | 0.009 | 0.009 |
| Bond angle (°) | 0.776 | 0.882 | 0.888 |
| **Validation** | | | |
| MolProbity score | 1.69 | 1.89 | 2.07 |
| Clashscore | 5.27 | 6.66 | 8.99 |
| Favored rotamers (%) | 99.89 | 99.37 | 99.63 |
| Poor rotamers (%) | 0 | 0.11 | 0 |
| **Ramachandran plot** | | | |
| Favored (%) | 93.84 | 91.02 | 88.29 |
| Allowed (%) | 6.16 | 8.98 | 11.68 |
| Disallowed (%) | 0 | 0.06 | 0.03 |

bonds, cation-π interactions, and long-range ionic interactions. Although the resolution of the map is not sufficient to specify these short-range interactions, we suggested possible interactions for reference (Figs. 2c, 3b, Supplementary Table 2). At the bifurcation point of the two stem structures, β'R77 locates like a wedge to separate G35* and G36* and forms a cation-π interaction with G35*. This cation-π interaction is often found between the terminal, exposed base of a nucleic acid bound to a protein and the protein loop that confines the nucleic acid. In addition, β'L78 and β'K79 are located between G35* and G36*, stabilizing the separation of stem I and stem II of the *put*RNA.

A mutant named *put*⁻, or mutant G, has the sequence A₄₃GAUC₄₇ and does not exhibit anti-termination activity[11]. Our transcription assay revealed that *put*⁻ also has poor anti-pausing activity (Fig. 1c). In the structure, this region does not directly interact with the RNAP, however, its counter-strand, from the 59th to 64th residues, forms a central area of the binding interface. Therefore, the base substitutions in the *put*⁻ mutant likely change the structure of the binding interface and disrupt *put*RNA binding to the RNAP. It is also possible that these mutations interfere with the proper folding of the *put*RNA as well.

To test the validity of the structure of the *put*EC modeled in the cryo-EM density, we introduced assorted mutations in the template DNA and performed in vitro radiolabeled transcription assays to examine the effects of the mutations on the anti-pausing activity (Fig. 3c, Supplementary Fig. 9). For the quantification of the anti-pausing activities of the mutants, the anti-pausing activities of wild-type *put* and *put*⁻ were set to 1 and 0, respectively, and the anti-pausing activity of each mutant was located on a linear scale accordingly (Details are in the Methods section). To display the location of the mutated residues as well their conservation, the conservation of the *put*RNA residues was calculated from the sequence alignment with ten known *put* sequences and marked by color (Fig. 3d, Supplementary Fig. 10a)[30,31]. Among the twenty-three mutations we generated, eleven mutants showed ≤ 20% anti-pausing activity (named 'inactivating' mutations) and three mutants showed ≥ 90% anti-pausing activity (named 'inert' mutations). The inactivating mutations, Δ3*−7*, U28*A, U28*C, G35*A, G35*U, G35*C, G45A*, A64*G, G35*C/A9*G, G35*A/A9*G, and G35*A/A9*U, suggest that (1) the 5′-region (from A3* to G7*) is essential for the anti-pausing activity. A₃GACG₇ and its base-pairing region, U₁₉CUGC₁₅ have relatively high conservation scores of (6,6,4,9,9) and (7,7,5,10,10), respectively. This region is the first RNA duplex formed during the *put*RNA synthesis, and therefore, may provide a platform for further RNA folding. (2) U28*, which protrudes toward the β'ZBD and binds to a small pocket is essential for the function. Interestingly, while U28*A and U28*C abolish the anti-pausing activity, U28*G retained ~60% of the activity. From the structure, we substituted the U28* with the other bases and found that G can form three hydrogen bonds with the surrounding β' residues while A and C form two and one potential hydrogen bonds, corroborating the result of the mutational study (Supplementary Fig. 10b). Interestingly, the original *put* residue, U28* forms fewer hydrogen bonds than guanine and adenine, but exhibits better activity than these, implying that U28* might have additional role(s) besides binding to the RNAP, or the mutants might have different structures from the modeled ones (Discussed below). (3) All of the G35* mutations we generated abrogated the anti-pausing activity of *put*RNA. We expected that the double mutants, G35*C/A9*G, and G35*A/A9*G might have some activity because they preserve the predicted base-pairing of G35*-A9* in the structure. However, mutating G35* to any base abrogated the anti-pausing activity and this was not recovered by the mutation of the base-pairing partner, implying that G35*, and possibly its base-pairing partner A9*, may have sequence-specific roles in the anti-pausing activity. We noticed that G35*U exhibited ~70% anti-termination activity in vivo[12]. This discrepancy could come from the different conditions encountered in vivo vs. in vitro. For example, the G35*U might form some intact or partially active *put*RNA in vivo, possibly aided by an unknown cellular factor(s) whereas in vitro synthesized *put*RNA containing G35U* could be inactive. (4) We also found that A64* is critical for the anti-pausing activity. This result is also consistent with the structural data because it contacts the stem I region of *put*RNA and the RNAP. All the inert mutations are of U20*, which lacks any significant interaction with other residues, supporting our structure. The remaining nine mutants exhibited moderate activities suggesting a significant, but not critical role of the residues (A8*, U21*, C25*, U32*, G43*). In summary, our mutagenesis study supports our cryo-EM structure of the *put*EC.

### The comparison of the *put*EC, the *put*-less EC, and other ECs

To determine if *put*RNA binding to the EC changes the conformation of the EC to suppress transcriptional pausing, we aligned the *put*EC with multiple EC structures including non-paused EC (PDB 6ALF), RNA hairpin-paused EC (PDB 6ASX), backtracked PEC (paused EC) (PDB 6RIP), and the *put*-less EC determined here (Fig. 4a, b, Supplementary Table 3)[21,32,33]. We assume that the *put*-less EC contains a roadblocked but unfolded RNA because (1) the majority

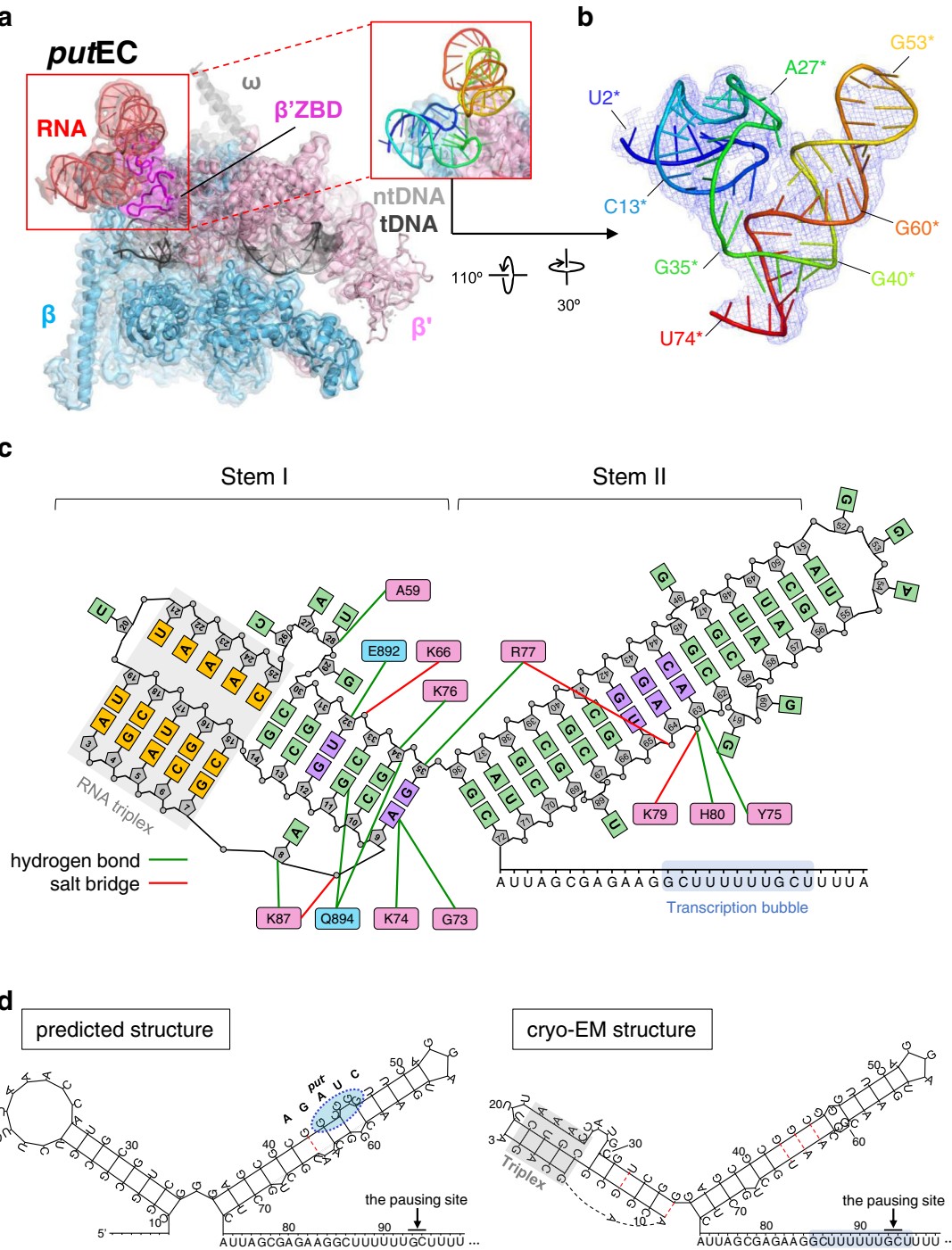

**Fig. 2 | The *put*EC structure. a** The cryo-EM map of the *put*EC (3.2 Å-resolution) is rendered as semi-transparent surface, colored as labeled, and superimposed with the *put*EC model. The RNAP domain that binds to *put*RNA, β'ZBD, is highlighted in magenta. The *put*RNA boxed in red is re-drawn in cartoon format and colored from blue to red according to the residue numbers. **b** The *put*RNA is drawn in an orientation revealing its overall structure more clearly. The *put*RNA figure contains 73 nucleotides from U2* to U74* (the first and the last base-pairing residues are A3* and C72*, respectively), and has 'V'-shape with two long stem-loop structures. The first stem region forms an RNA triplex. **c** A schematic diagram of the *put*RNA structure and its potential interactions with RNAP β' and β subunits. The *put*RNA bases forming canonical WC base pairs and the unpaired bases are colored in light green. The bases forming non-canonical base pairs are colored in light purple. The

RNA triplex region is marked in gray color, and the bases in the triplex are colored in orange. The pink and blue boxes represent β' and β residues, respectively, and the lines between the RNAP residues and the *put*RNA nucleotides indicate potential hydrogen bond or salt bridge interactions. The ribose, shown as pentagon, and the phosphates, shown as circles, are colored in gray and the residue numbers of the *put*RNA are written in the ribose. Transcription bubble is marked with blue shade. **d** (Left) The predicted *put*RNA structure[11]. The pausing site and the *put*⁻ mutation that abolishes the anti-termination activity are marked. Non-canonical base pairs are represented with red dotted line. (Right) The *put*RNA structure modeled based on the cryo-EM map. Triple helix region is marked by a gray box. Non-canonical base pairs are represented with red dotted lines. Transcription bubble region and the pausing site are marked.

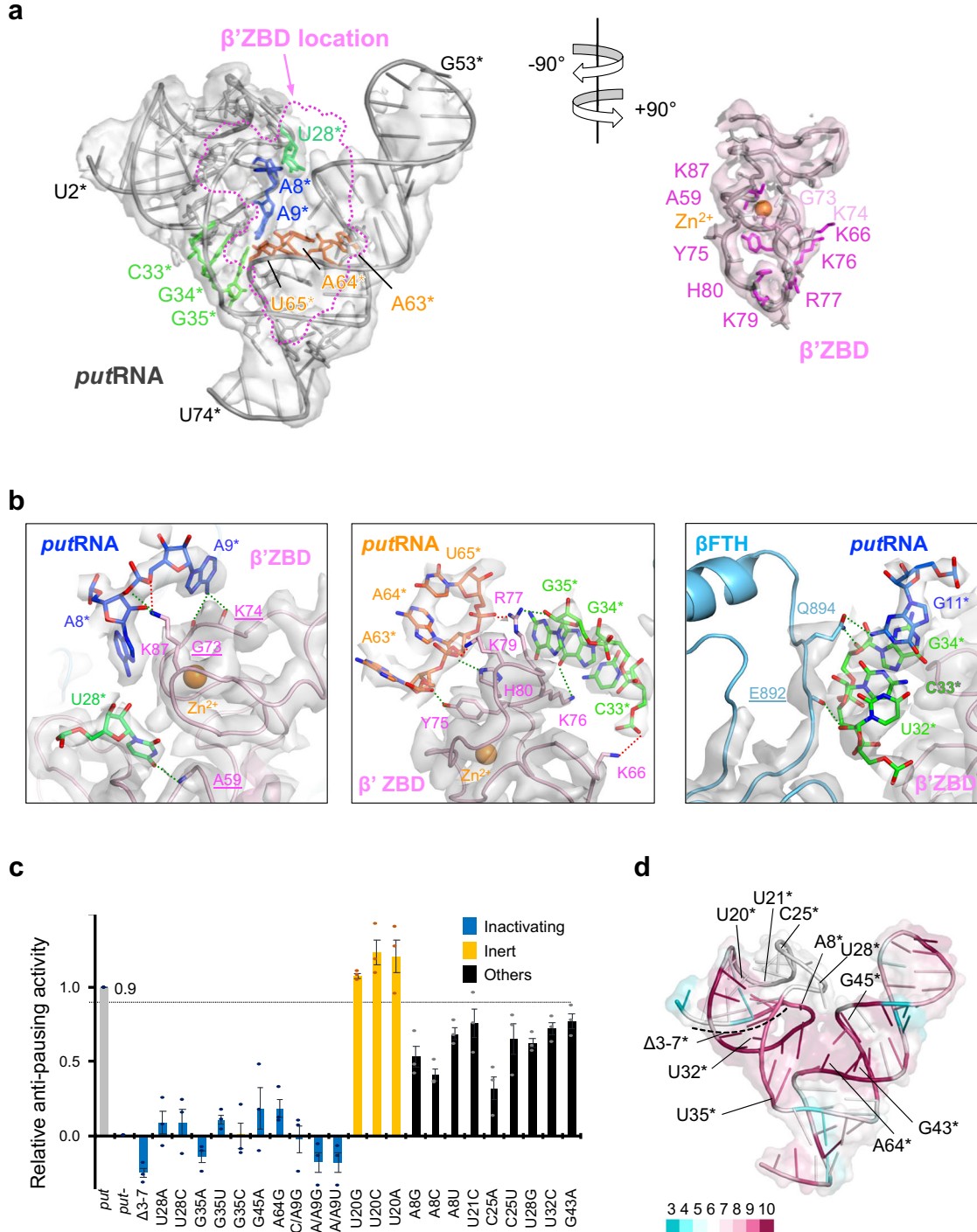

**Fig. 3 | Interaction between the *put*RNA and the RNAP. a** The binding interface between the *put*RNA and β'ZBD is split and each region is rotated by 90° in the opposite direction to display the binding surface. The cryo-EM map is rendered as semi-transparent surface, colored as labeled, and superimposed with the structure. The RNA and amino acid residues in contact with the other partner are drawn in stick format. The amino acid residues forming polar interactions via side chains are colored and labeled in magenta while the ones interacting via peptide backbone are in light pink (detailed in **b**). The β'ZBD binding region mapped onto the *put*RNA is marked by dotted line. **b** (Left two boxes) Close-up views of the *put*RNA-β'ZBD interface with potential interactions - salt bridges (red dotted line) and hydrogen bonds (green dotted line). The labels of the amino acids forming hydrogen bonds via backbone peptides are underlined on their label. The *put*RNA residues are colored according to their residue numbers. (Right box) Close-up view of the *put*RNA-β

flap-tip helix. The cryo-EM density is superimposed on the model in all three boxes. **c** Measurement of the anti-pausing activities of the *put*RNA mutants. From the radiolabeled transcription assay, the relative anti-pausing activity of each mutant is calculated and plotted (details in Methods). The mutants were classified into three groups according to the activities and labeled as 'Inactivating' (<20%), 'Inert' (>90%), and 'Others' (between 20% and 90%). The relative activity of 0.9, which separates the inert and other mutations, is marked by a dotted line. The assays were done in triplicates (*n* = 3 independent experiments), and data are presented as mean values ± SEM. All data points are plotted as dots in different colors for clarity. **d** Conservation of *put*RNA residues and the location of the *put*RNA mutations in **c**. The sequence alignment is done in LocARNA server[30]. The conservation score in each residue is measured from the alignment and colored on the structure as indicated in the color bar in the figure. Source data are available as a Source Data file.

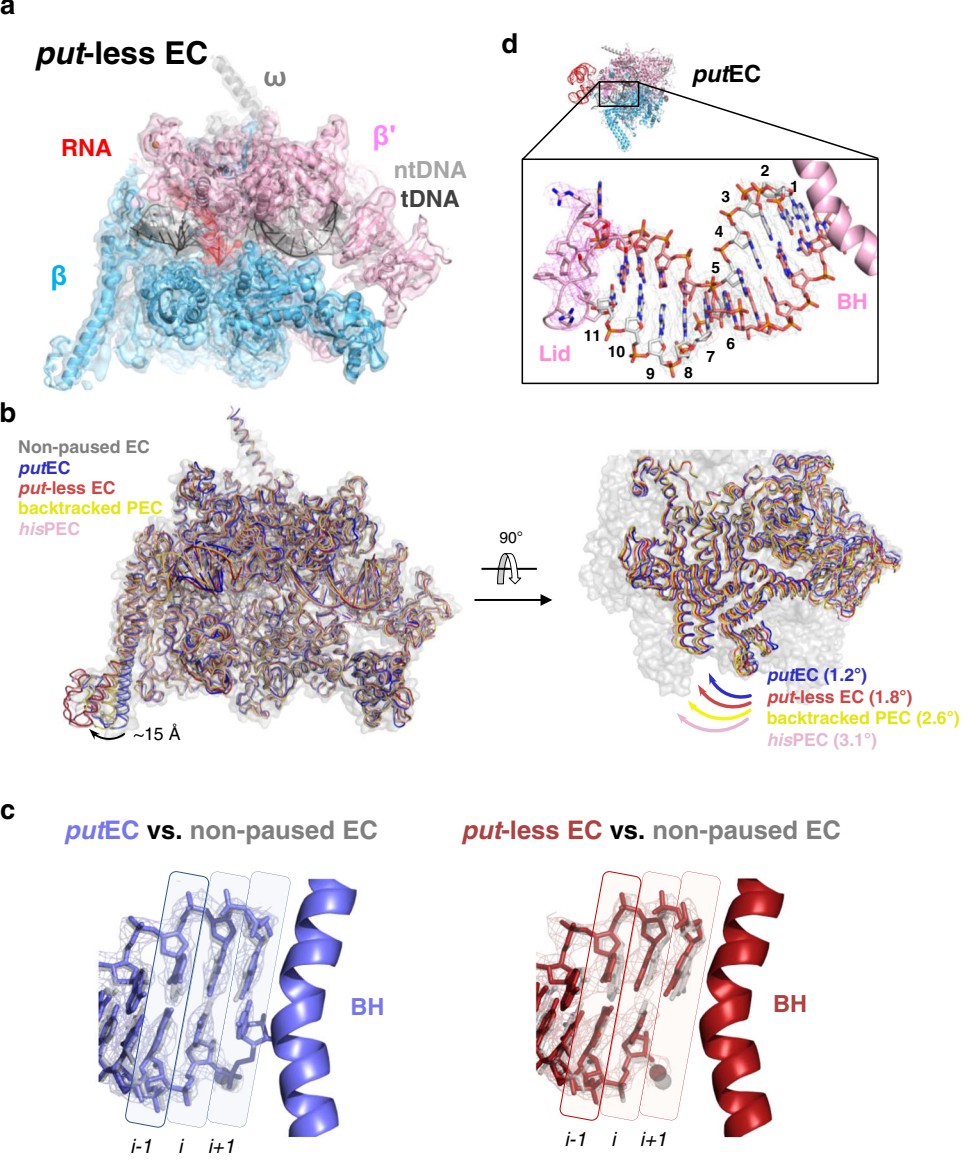

**Fig. 4 | Comparison of *put*EC, *put*-less EC, and other ECs. a** The cryo-EM map of the *put*-less EC (3.6 Å) is rendered as semi-transparent surface, colored as labeled, and superimposed with the final *put*-less EC model. Different with the *put*EC map, no structured RNA density was observed except that of an RNA-DNA hybrid. **b** The swiveling of the *put*EC and the *put*-less EC are compared with that of other representative ECs containing non-paused EC (PDB 6ALF), backtracked PEC (swiveled, PDB 6RIP), and *his*PEC (PDB 6ASX)[21,32,33]. All five ECs were aligned according to the core module. (Left) Front view of the aligned ECs. Non-paused EC is drawn in semi-transparent surface and loop formats in gray color. Other ECs are drawn in loop format and colored as labeled. βSI2 domains show distinct conformations between non-paused EC/*put*EC and *put*-less EC/*his*PEC, with ~15 Å displacement. βSI2 of the backtracked PEC is located in the middle of the two groups. (Right) Top view. Only

the swivel modules (clamp, shelf, jaw, SI3 domains and C-terminal region of β' subunit) of the aligned ECs are drawn for clarity. The swiveling angles relative to the non-paused EC are shown. **c** Comparison of the RNA-DNA hybrids near RNAP active site with the *put*EC, the *put*-less EC, and the post-translocated, non-paused EC (PDB 6ALF)[21]. The RNA-DNA hybrid, bridge helix (BH), and Mg²⁺ ions are colored as labeled. The cryo-EM densities of the *put*EC and *put*-less EC hybrids are shown in mesh, and their Mg²⁺ ions are shown as semi-transparent spheres. The base-pair locations are marked with background boxes, and the BH beside the *i + 1* site is drawn for reference. **d** The *put*EC contains 11-bp RNA-DNA, one base pair longer than other reported bacterial EC structures. The lid and bridge helix (BH) of β' are colored in light pink, and each nucleotide is numbered from downstream to upstream template DNA.

(>~70%) of the sample was roadblocked properly (Fig. 1d), and both the *put*EC and the *put*-less EC together comprise ~60% of the EC population in the cryo-EM data, (2) the third EC class, σ70-bound *put*EC shows extra RNA duplex density connected to the *put*RNA suggesting that this class was not properly roadblocked, and (3) the *put*-less EC map contains some weak RNA density around the RNA exit channel and the β'ZBD, implying that the RNA is present, but it is not well-structured. We suggest this *put*-less EC could serve as a good negative-control model as shown in a previous study[32].

We first examined the swiveled states of the ECs (Fig. 4b). Swiveling indicates the rigid-body rotation of a set of domains–the clamp, dock, shelf, jaw, SI3, and the C-terminal region of the β' subunit–about an axis parallel to the bridge helix toward the RNA exit channel, and known to interfere with the proper folding of the trigger-loop which is required for efficient nucleotide addition to the nascent RNA. Swiveling was first introduced from the structural study of *his*PEC, and later revealed in the backtracked PEC, implying that the swiveling motion potentially plays an important role in both RNA hairpin pause and backtrack pause[32–34]. The alignment of the EC structures according to

the core module revealed that the *put*EC structure is most similar to the non-paused, active EC conformation, having the lowest RMSD values between $C_\alpha$-carbons of domains as well as the smallest swiveling angle of 1.2° (Fig. 4b, Supplementary Table 3). The *put*-less EC is more swiveled than the *put*EC, having a swivel angle of 1.8°, although the swiveling angle of the *put*-less EC was less than that of the *his*PEC or backtracked PEC (3.1° and 2.6°, respectively; Supplementary Table 3). Interestingly, the conformational difference between the *put*EC and *put*-less EC is more noticeable in the βSI2 (or βi9) region with 15 Å-distance between the $C_\alpha$ atoms of βE1006, which is located at the end of the βSI2 domain. While the swiveling motions of the aligned ECs are relatively continuous with the rotation angles from 1.2° to 3.1°, the arrangement of the βSI2 is more discrete – the βSI2 in the *put*EC overlaps with that of the non-paused EC while the βSI2 of the *put*-less EC is in the same location with the *his*PEC. Interestingly, the βSI2 of the backtracked PEC is located between the two conformations. These conformational features suggest that the proper folding and binding of the *put*RNA to the EC moved the EC toward the non-swiveled, active state, aiding pause escape or omission.

The strength of the RNA-DNA hybrid influences pausing and termination[35,36]. Therefore, we compared the RNA-DNA hybrid of the *put*EC and the *put*-less EC (Fig. 4c, left). In the *put*EC, the active site region of the RNA-DNA hybrid exhibited a post-translocated state similar to the non-paused EC at the high threshold value of the map. As the threshold value decreases, the *put*EC map revealed a density blob for a nucleotide base that base pairs with the template DNA base at the *i + 1* site. This density became connected to the nascent RNA at the lower density threshold. As stated above, we suspect that this results from the mixed population of the nascent RNAs roadblocked at either +94 or +95 position, having either post- or pre-translocated states, respectively. However, we did not observe any classes having a folded trigger-loop with the SI3 domain shifted closer to the βlobe domain as in the *Eco* RNAP structure of the pre-translocated state[24]. In addition, the *put*EC contained 11 template DNA bases in the RNA-DNA hybrid, in contrast to other reported EC structures (Fig. 4d). To contain one additional nucleotide in the main channel, the lid, which is known to aid the unwinding of the RNA-DNA hybrid, is pushed by about 2.6 Å (by the $C_\alpha$ atom of β'256D) compared to the known non-paused EC (Supplementary Fig. 11a)[19,21,33]. However, it is not certain if this 11-nt hybrid is just an alternative conformation of an EC, or a specific conformation in the *put*EC.

The *put*-less EC showed distinct RNA-DNA base-pairing at the *i + 1* site (Fig. 4c, right). In the *put*-less EC, the template DNA base at the *i + 1* site is more tilted toward the RNA base at the *i* site; therefore, it is not optimally placed for substrate binding. In fact, the RNA base at the *i* site is more closely associated with the DNA base at the *i + 1* site than that of the *i* site. Consequently, the base-pairing hydrogen bonds are broken between the template DNA base and the nascent RNA base at the *i* site. The remaining region of the RNA-DNA hybrid of the *put*-less EC overlaps well with that of the non-paused, active EC as in the *put*EC. The conformational difference of the nucleotides at the active site between the *put*EC and the *put*-less EC indicates that *put*RNA binding to the β'ZBD influences the active site conformation, even though the catalytic magnesium ion is ~62.5 Å away from the zinc ion in the β'ZBD. This was also shown in the *his*PEC structure, where the pause hairpin placed in the RNA exit channel has an influence on the active site as well as the bridge helix[32,34,37]. The length of the template DNA in the RNA-DNA hybrid of the *put*-less EC was also 11-nt, implying that this longer RNA-DNA hybrid is not caused by the *put*RNA.

In addition to these changes, we also observed that the RNAP domains of the *put*EC have similar locations to those of the non-paused EC while the domains in the *put*-less EC have a similar arrangement with those of backtracked PEC (Supplementary Table 4). Although we could not find any density for the backtracked RNA in the *put*-less EC, the pausing site was expected to have a backtrack pause. In summary,

from the structures of the *put*EC, *put*-less EC, and other ECs, we found that the *put*RNA binding to the EC leads to the anti-pausing activity by promoting the active, non-swiveled conformation of the EC.

## σ[70]-bound *put*EC structure

The third EC population, σ[70]-bound *put*EC, contains a σ[70] bound to the clamp helices in addition to the well-folded *put*RNA as in the *put*EC (Fig. 5a, Supplementary Fig. 2). In contrast to the holoenzyme structure, the σ[70]-bound *put*EC map reveals only $\sigma_{1.2}$, $\sigma_{NCR}$ and a part of $\sigma_2$, indicating that the $\sigma_2$ binding to the EC is relatively stable while the other σ domains are very mobile as predicted in a prior study[38]. It has been reported that σ[70] can remain associated with RNAP after promoter escape and the association is enhanced when the non-template DNA contains a −10 element-like sequence in the promoter-proximal region that induces σ-dependent pausing[39–41]. In particular, σ-dependent pausing provides a time and space window for the anti-termination λQ protein to bind to the EC and read through the intrinsic terminator[42,43]. Recently, cryo-EM structures of σ[70]-bound ECs were reported in the context of 21Q-, λP$_R$'-, and Qλ-associated ECs[44–48]. While these complexes are at the paused state in that the $\sigma_2$ domain interacts with a −10-like sequence, our σ[70]-bound *put*EC is not in a σ-dependent paused state and contains > 100 base-long RNA having a σ[70] in a different conformation from those in other σ[70]-bound ECs (Fig. 1d).

In the σ[70]-bound *put*EC structure, we noticed that the RNAP contains an open clamp (79.3 Å opening), which is ~20 Å larger than the non-paused EC[23]. This suggests that the σ[70]-bound *put*EC is in an inactive state. We suspect that this class might represent the partial run-off EC population that appeared in the nMS analysis (Fig. 1d) because (1) the main channel of the RNAP did not contain downstream duplex DNA while the RNA-DNA hybrid was present and (2) an RNA duplex density, which is connected to both *put*RNA and the RNA-DNA hybrid, was observed in the RNA exit channel, indicating that the RNA was transcribed beyond the roadblock site (Fig. 5a, Supplementary Fig. 11b). We used the RNAfold Server to search potential RNA secondary structures in the template DNA and found that it contains a potential RNA hairpin sequence downstream of the roadblock site (Supplementary Fig. 11b)[49]. We, therefore, modeled the RNAP and nucleic acid scaffold into the map and found that the potential RNA hairpin matches well with the extra density observed in the RNA exit channel (Fig. 5a, Supplementary Fig. 11b). The location of this extra RNA duplex overlaps with the pause hairpin in the *his*PEC[32,34].

The *put*RNA density in the σ[70]-bound *put*EC was at a lower resolution than that in the *put*EC; however, the *put*RNA map region was identical to that in the *put*EC. The RMSD of the whole atoms in the *put*RNA region in the σ[70]-bound *put*EC and the *put*EC was only 0.839 Å. To compare the $\sigma_2$-RNAP interaction in the initiation and the elongation stages, we aligned the RNAP clamp-$\sigma_2$ domain regions from the σ[70]-bound *put*EC and the recently published RPo (RNAP-promoter open complex) structure[25]. For the σ[70]-bound *put*EC, we only modeled the visible part for the σ[70] (σ[70] residues 112–151 and 214–447). Then, we compared the two structures only via the modeled σ[70] regions and other σ domains were excluded in the comparison discussed below. Not surprisingly, the binding interface between the σ[70] and the RNAP, in particular, the β'clamp domain, was different between the RPo and the σ[70]-bound *put*EC (Fig. 5b). The binding interface between the β' subunit and the σ[70] was 812 Å² in the RPo and the interface mostly occurs on the β'clamp helices. By contrast, in the σ[70]-bound *put*EC, the interface area was 1287 Å². This unexpected increase in the binding area results from the newly-formed interface between β'-clamp-toe domain (ranging 144–179)[50] and the σ[70]$_{NCR}$, the non-conserved σ[70] region between $\sigma_{1.2}$ and $\sigma_{2.1}$ (ranging 274–307 and 359–374 in the structure, Fig. 5b) that does not participate in the RNAP-σ[70] interface in the RPo. Since both β'-clamp-toe and σ[70]$_{NCR}$ are conserved in the γ-proteobacteria, the interaction between these two domains might be specific for the bacteria class. In addition, the shifted position of the $\sigma_2$

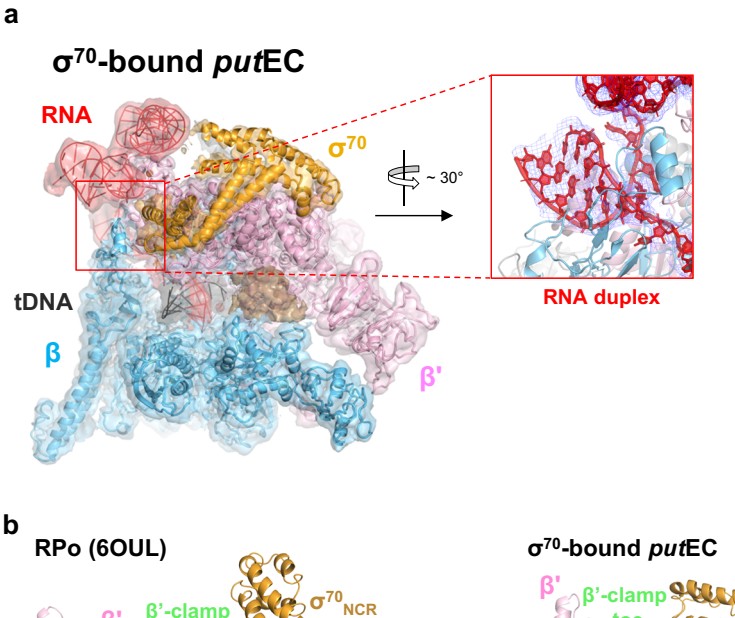

**Fig. 5 | The σ70-bound *put*EC structure. a** The cryo-EM map of the σ70-bound *put*EC (3.6 Å) is rendered as semi-transparent surface, colored as labeled, and super-imposed with the final σ70-bound *put*EC model. The unexpected RNA hairpin found in the RNA exit channel is zoomed, rotated by -30°, and re-drawn with the density in a red box. The downstream duplex DNA was not observed in the main channel. Instead, an unknown blob, colored in light brown, was present in the location. **b** Comparison of the σ70-RNAP binding between the RPo and the σ70-bound *put*EC. Nucleic acids were not drawn for clarity. β' clamp helices (β'CH), the main structural element interacting with σ70 in the RPo, are highlighted in magenta color. σ70 is colored by domains as labeled. β'-clamp-toe, which interacts with σ_NCR in the σ70-bound *put*EC, is colored green.

domain in the σ70-bound *put*EC is more suitable for the σ70 to associate with the progressing EC because this conformation provides space for the upstream DNA to rewind and exit from the main channel of the RNAP. If the σ70 is bound to the RNAP as in the holoenzyme, σ70 would clash with the exiting upstream duplex DNA. However, at the moment, further investigation would be required to see whether these new interactions between the σ70 and the RNAP in the σ70-bound *put*EC are due to the transcription stage transition from initiation to elongation, or to the clamp opening which inactivates the transcription activity of the RNAP.

Additionally, we found a low-resolution blob in the main channel for the downstream DNA (Fig. 5a). The DNA scaffold used in the study spans to +122 position while the RNA modeled in this map ends at +105. nMS analysis revealed three RNA populations of 110-mer, 114-mer and 116-mer (Fig. 1d). Therefore, there should be some downstream duplex DNA around the RNAP. However, the low-resolution of the blob pre-vents us from locating any specific molecule in the density. We suspect that the blob could be either from the downstream duplex DNA, which is very mobile due to the open clamp conformation, or from the σ70_{1.1} because the σ70_{1.1} is known to bind at the position in the holoenzyme before the enzyme binds to promoter DNA. We would need further investigation to confirm this speculation.

## Discussion

In this study, we extended prior studies on the *put*RNA by determining its three-dimensional structure when complexed with RNAP. Our result corroborates previous analyses suggesting a two-stem structure with multiple indents and bulges. However, cryo-EM structures also revealed new and unexpected features such as an unexpected boundary of the *put* transcript, a short triple RNA helix in the *put*L stem I, and alternative base pairs. The importance of many of these features is strengthened by the observed effect of specific *put* mutations[11,12]. The structure provides clear physical evidence that the *put*RNA binds to the β'ZBD, a result that is strongly supported by prior genetic and biochemical experiments on *put*RNA. The structure also revealed a mechanistic explanation for the anti-pausing activity promoted by *put*RNA-RNAP interaction. When *put*RNA is bound to the β'ZBD, the RNAP is held in a non-swiveled, active conformation, which is asso-ciated with anti-pausing activity as previously shown in the RfaH-associated EC[19]. In contrast, a *put*-less EC exhibited a swiveled con-formation suggesting that the EC is in a paused state when transcrip-tion elongation is physically blocked at a pausing site. Together, the structures revealed that *put*RNA promotes RNA synthesis by resisting swiveling.

We were surprised to observe a *put*EC population that retained σ70 even though the EC had progressed about 100 nucleotides from the start of transcription. The occurrence of the σ70-bound *put*EC and its structure suggests a few intriguing points. First, the σ70-bound EC successfully folded *put*RNA, even more efficiently than a complex lacking σ70. We found that the ratio between the *put*EC and the *put*-less EC is roughly 2:3 from the number of particles in each class, presumably reflecting the success rate of *put*RNA folding in vitro. Curiously, there was no *put*-less σ70-bound EC, suggesting that the presence of σ70 aided the proper folding and stabilization of

*put*RNA. We found that the *put*L sequence contains a weak −10-like sequence (NANNAT) located at positions +23 to +28 relative to the start of the *put*L transcript, which lies on the third strand of RNA triple helix and a bulge region of stem I (Figs. 1a, 2b, c). A −10-like sequence is known to induce σ-dependent pausing by engaging its non-template DNA region with the $\sigma_2$ domain[51]. We suggest that this sequence may cause σ-dependent pausing which facilitates *put*L folding by providing more time. Notably, among the ten *put*RNA sequences we aligned, all the *put*Ls contain the identical −10-like sequences while all *put*Rs do not (Supplementary Figs. 10a, 11c). Therefore, we speculate that σ-dependent pausing may be necessary for *put*L folding but not for *put*R which is located further downstream of its promoter. In addition, U28* is completely conserved in *put*L and the critical 6th residue of the −10-like sequence. Although U28*G exhibited intermediate activity in our mutagenesis study, U28*A and U28*C nearly abolished activity, supporting the existence and importance of σ-dependent pausing at this position. Furthermore, we examined the ratio between $\sigma^{70}$-bound EC and $\sigma^{70}$-unbound EC from both the *put*EC sample and the *put*−-EC to see if the presence of *put* affects the $\sigma^{70}$ retention (Supplementary Fig. 12). The *put*−-EC was prepared in exactly the same way as the *put*EC preparation except the *put*− template was used as the DNA scaffold and did not show any well-folded *put*RNA density in the cryo-EM maps. From the cryo-EM data analysis, the percentages of $\sigma^{70}$-bound EC in the *put*EC (having intact *put*) and the *put*−-EC sample were ~40.7% and ~44.2%, respectively, suggesting that the presence of *put* does not affect $\sigma^{70}$ retention. Second, the $\sigma^{70}$-bound EC was resistant to the LacI roadblock. The $\sigma^{70}$-bound *put*EC revealed an extra density for a duplex RNA in the RNA exit channel, suggesting that the retained $\sigma^{70}$ modified the EC to overcome the roadblock during elongation (Figs. 1d, 5a, Supplementary Fig. 11b).

Structural studies on prokaryotic anti-termination complexes including λN, Q21, *Xoo* P7, Qλ, and HK022 *put* suggest general strategies for anti-termination[7,44,45,47,48]. (1) The anti-termination factors inhibit RNA hairpin formation by either narrowing the channel or hindering the RNA hairpin folding (Supplementary Figs. 4, 13). The RNA exit channel is thought to aid RNA hairpin formation by its positively-charged residues located inside the channel[32]. In Q21, Qλ and *Xoo* P7 anti-termination complex, the anti-termination factors, Q21, Qλ, and P7 proteins bind at the mouth of the RNA exit channel and confine the channel (Supplementary Fig. 13). The narrowed RNA exit channel only allows single-stranded RNA to move through it and restricts nascent RNA folding for hairpin-dependent pausing and intrinsic termination. In λN anti-termination complex, λN binding to the EC remodels the bound NusA and NusE to destabilize the RNA hairpin folding. In addition, the rearranged Nus factors bind to β flap-tip, which stabilizes RNA hairpin pause, possibly preventing the flap-tip from assisting RNA hairpin pausing and termination[52]. Like λN, HK022 *put* also does not narrow the RNA exit channel directly. Instead, the phosphate backbone of the *put*RNA is located near the RNA exit channel, prohibiting the RNA hairpin formation with its negative-charged surface. Modeling an RNA duplex in the RNA exit channel of *put*EC shows that the phosphate backbones of the modeled RNA duplex and the *put*RNA are just ~5 Å apart from each other (Supplementary Fig. 4). In addition, the β flap-tip binds to the *put*RNA, possibly sequestering it from assisting RNA hairpin pause as in the λN-anti-termination complex. (2) In general, anti-termination proteins stabilize the elongation-proficient conformation of EC. λN transverses the RNAP hybrid cavity stabilizing the active form of the EC and binds to the upstream duplex DNA, enhancing the anti-backtracking and anti-swiveling activity of NusG. In the Q21-EC structure, Q21 binding is not compatible with swiveled conformation. Therefore, Q21 counteracts swiveling, leading to anti-pausing[47]. Our data suggest that *put*RNA also reduces swiveling. This stabilization of the active form of an EC may consolidate the RNA exit channel so that it can no longer accommodate the folding of secondary structures that promote pausing and termination[53–55].

Komissarova et al.[17], found that ΔU68* does not suppress termination, but retains anti-pausing activity in vitro. U68* is located at the lower region of stem II like a wedge, forming no base-pairing. According to our modeling, the presence of U68* kinks the stem II ~19° (Supplementary Fig. 14). This perturbation might weaken the stability of the *put*RNA folding by widening the space between the two stem-loop structures. In addition, the structural change would affect the interface between the *put*RNA and the RNAP because the interface is composed of *put*RNA residues from both stem I and II. Therefore, *put*RNA without U68* might be well-folded and reduces pausing immediately after synthesis but may unfold or dissociate from the RNAP before encountering terminators located further downstream. Alternatively, the mutant RNA may not be able to adopt an anti-terminating structure which could be different from the anti-pausing structure in vitro.

In λ phage paradigm, the anti-termination factor λN plays the role of gatekeeper for the infection process. In other words, λN accumulation is required to transcribe early genes of the genome. HK022, instead, has Nun protein, which competes with λN and blocks λ transcription. In addition, the HK022 genome harbors the *put* element in the place for the λ *nut* (N-utilization) sites, which are required for the action of λN. By substituting the N protein with Nun, HK022 acquired immunity against its competitor, λ. HK022, instead, lacks a λN-like anti-termination factor, but relies solely on the *put*RNA to promote full expression of its early genes. These differences benefit HK022 survival, without increasing transcription regulation complexity.

In this study, we investigated the anti-pausing mechanism of *put*RNA. Since transcriptional pausing is a prerequisite of transcriptional termination, our results provide important insights into the mechanism of *put*RNA action. It remains possible that *put*RNA may adopt different structures and/or interactions with RNAP to promote anti-termination as prior studies indicate that anti-pausing and anti-termination activities differ. To deepen our understanding of these events, structural studies on the *put*RNA-associated EC at a terminator sequence would be required.

## Methods

### Protein expression and purification

*Eco* RNAP was prepared as described previously[25,56]. Briefly, pET-based plasmid that contains *rpoA* (α), *rpoB* (β), *rpoC* (β′) with C-terminal deca-histidine tag and *rpoZ* (ω) was co-expressed with a pACYCDuet-1 plasmid contained *rpoZ* in BL21(DE3) (Novagen). The cells were grown at 37 °C in the LB broth media in the presence of 100 μg/mL Ampicillin and 34 μg/mL Chloramphenicol, and transferred to 30 °C when the $OD_{600}$ reached 0.3. Protein expression was induced at an $OD_{600}$ of 0.6–0.8 with 1 mM IPTG for 4 hours. Cells were harvested, resuspended in lysis buffer (50 mM Tris pH 8.0, 5% glycerol, 1 mM EDTA (pH 8.0), 1 mM $ZnCl_2$, 10 mM DTT, home-made protease inhibitor cocktail), and lysed by French Press (Avestin) at 4 °C. The lysate was precipitated by adding polyethyleneimine (PEI, 10% (w/v), Sigma Aldrich) to a final concentration of 0.6% (w/v) dropwise. The pellets were washed three times with wash buffer containing TGED (10 mM Tris pH 8.0, 5% glycerol, 0.1% EDTA pH 8.0, 10 mM DTT) + 0.5 M NaCl, and the RNAP was eluted from the pellet with elution buffer (TGED + 1 M NaCl). The eluted RNAP was precipitated by adding ammonium sulfate (35 g per 100 ml solution) and eluted again with chelating buffer (20 mM Tris pH 8.0, 1 M NaCl, 5% glycerol) to be loaded onto Hitrap IMAC HP columns (Cytiva) for purification by $Ni^{2+}$-affinity chromatography. The pulled protein by adding imidazole gradient was dialyzed in TGED + 100 mM NaCl buffer and loaded onto a Biorex-70 column (Bio-rad) for ion exchange chromatography. Eluted RNAP by NaCl gradient was concentrated by Amicon Ultra centrifugal filter (Merck Millipore), and loaded onto HiLoad 16/600 Superdex 200 pg column (Cytiva)

equilibrated SEC buffer (TGED + 0.5 M NaCl) for size-exclusion chromatography. The purified protein was supplemented by 15% glycerol, flash-frozen in liquid nitrogen, and stored at −80 °C until use.

Full-length *Eco* σ[70] was expressed from pET21-based expression vector encoding an N-terminal hexa-histidine tag followed by a PreScission protease (GE healthcare) cleavage site. The full-length *Eco* σ[70] plasmid was transformed BL21(DE3) cells and grown at 37 °C. Protein expression was induced at an $OD_{600}$ of 0.7 with 1 mM IPTG and incubated for 4 hours at 30 °C. Cells were harvested, resuspended in σ[70] lysis buffer (20 mM Tris pH 8.0, 500 mM NaCl, 5% Glycerol, 5 mM Imidazole, home-made protease inhibitor cocktail) and lysed by French Press. The supernatant was loaded to Hitrap IMAC HP column (Cytiva) equilibrated with 20 mM Tris pH 8.0, 500 mM NaCl, 5% glycerol. The eluted protein by adding imidazole gradient was concentrated using Amicon Ultra centrifugal filter (Merck Millipore) and injected to HiLoad 16/600 Superdex 200 pg (Cytiva) equilibrated in TGED + 500 mM NaCl. The final elution was flash-frozen using liquid nitrogen after adding 15% glycerol.

Lac repressor (LacI) was purified as described previously[57]. LacI-containing pBAD plasmid with Kanamycin resistance (pBAD_Kan-LacI) was obtained from Addgene (plasmid #79826). BL21(DE3) cells that were transformed with the plasmid were grown overnight at 37 °C in 2X YT media containing 50 µg/mL Kanamycin. The seed culture was added to 2× YT media containing 50 µg/mL Kanamycin at 1:100 ratio, grown at 32 °C for 2 hours, and moved to 16 °C. Protein expression was induced with 0.2% L-arabinose for 16 hours incubation right after changing the temperature to 16 °C. Cells were harvested and lysed by French Press in lysis buffer (50 mM sodium phosphate buffer pH 8.0, 500 mM NaCl, 20 mM Imidazole, 2.5% glycerol, 1 mM DTT, 10 mM $MgCl_2$, 0.1% Tween-20, 1 mg/mL lysozyme, home-made protease inhibitor cocktail). The lysate was added by 1000 U of DNaseI, and centrifuged to remove cell debris. The supernatant was loaded onto Hitrap IMAC HP (Cytiva) that pre-equilibrated with 50 mM sodium phosphate buffer (pH 8.0), 500 mM NaCl, 20 mM imidazole, 2.5% glycerol, and 0.2 mM DTT. Protein was eluted with 20 mM sodium phosphate buffer (pH 7.4), 300 mM NaCl, imidazole gradient from 30 to 300 mM and concentrated using 30 K MWCO Amicon Ultra Centrifugal Filter (Merck Millipore). The concentrated protein was injected onto HiLoad 16/600 Superdex 200 pg (Cytiva) gel filtration column equilibrated with 20 mM Tris-HCl (pH 8.0), 150 mM KCl, 5 mM $MgCl_2$, and 1 mM DTT. The final eluted protein was added by 15% glycerol, flash-frozen, and stored at −80 °C until use.

## Radiolabeled in vitro transcription assay

In vitro transcription assay is performed as described previously[58]. Holoenzyme was reconstituted by mixing *Eco* RNAP and *Eco* σ[70] with 1:2 molar ratio, and incubating for 15 min at 37 °C. Holoenzyme and DNA were mixed with 4:1 molar ratio in glutamate-based T buffer (20 mM Tris-glutamate pH 8.0, 10 mM Mg-glutamate, 150 mM K-glutamate, 5 mM DTT), and incubated at 37 °C for 10 min to make RPo. RPo and LacI were mixed with 1:10 molar ratio and incubated at 37 °C for 10 min. The final concentration of holoenzyme and template DNA in the reaction mixture was 50 nM and 12 nM, respectively. Transcription was started by adding rNTP mix to final concentrations of 200 µM ATP, 200 µM UTP, 200 µM GTP, 25 µM CTP (Cytiva) and 0.05 µM α-[32]P-CTP (PerkinElmer) at 37 °C, and quenched after 2 min by adding 2× loading buffer (10 M Urea, 50 mM EDTA pH 8.0, 0.05% bromophenol blue, 0.05% xylene cyanol). To show the roadblocked EC is capable of further transcription, the roadblocked EC was added by 2 mM IPTG, incubated for 2 min at 37 °C for LacI dissociation, and added additional rNTP to final concentrations of 162 µM ATP, 162 µM UTP, 162 µM GTP, 75 µM ATP and 0.15 µM α-[32]P CTP. The samples were loaded on 10% Urea-PAGE gel and ran in 1X TBE. The gel was exposed to an imaging plate (Fujifilm) for 2 hr, and the imaging plate was scanned to get an image (Typhoon™ FLA 7000).

For the mutational study, 50 nM holoenzyme and 12 nM template DNA were used for the transcription assay without roadblocking. In addition, the transcription reaction was quenched at 0-, 0.5-, and 2-min time point, and the data at 0.5 min were used to estimate the relative anti-pausing activity plotted in Fig. 3c although using 2-min data also showed similar result (data not shown). For the transcription reaction, 200 µM ATP, 200 µM UTP, 200 µM GTP, 25 µM CTP, and 0.05 µM α-[32]P CTP were used. For the estimation of the relative anti-pausing activity, we measured the intensities of the paused and the run-off transcripts of the *put* constructs, and calculated the fraction of the paused transcripts by dividing the intensity of the paused transcript by the sum of the intensities of the paused and run-off transcripts (Supplementary Fig. 9). The fractions of the paused transcripts were calculated for the wild-type *put*, inactive *put*⁻, and mutant *put* constructs, and their relative anti-pausing activities were calculated by the equation below and plotted:

$$\text{Relative anti-pausing activity of mutant x} = 1 - \frac{(P_X - P_{WT})}{(P_{put-} - P_{WT})}$$

$$(P_X = \text{the fraction of the paused band of mutant x})$$

For the paused fraction quantification, the intensities for the run-off and paused transcripts were calibrated according to the number of cytosines the transcripts contain. The assay was done in triplicate ($n = 3$ independent experiments).

## Native mass spectrometry analysis

The RNA portion of the de novo reconstituted *put*EC was prepared by phenol/chloroform extraction, resuspended in RNase-free water and flash-frozen in liquid nitrogen. Prior to analysis, the sample was thawed and then buffer-exchanged into nMS solution (500 mM ammonium acetate, 0.01% Tween-20, pH 7.5) using Zeba desalting microspin columns (Thermo Fisher). The buffer-exchanged sample was diluted to 5 µM with nMS solution and was loaded into a gold-coated quartz capillary tip that was prepared in-house. The sample was then electrosprayed into an Exactive Plus EMR instrument (Thermo Fisher Scientific) using a modified static nanospray source[59]. The MS parameters used were similar from previous work[22]: spray voltage, 1.2 kV; capillary temperature, 150 °C; S-lens RF level, 200; resolving power, 8750 at $m/z$ of 200; AGC target, $1 \times 10^6$; number of microscans, 5; maximum injection time, 200 ms; in-source dissociation, 10 V; injection flatapole, 10 V; interflatapole, 7 V; bent flatapole, 6 V; high energy collision dissociation, 85 V; ultrahigh vacuum pressure, $6.6 \times 10^{-10}$ mbar; total number of scans, 100. Mass calibration in positive EMR mode was performed using cesium iodide. Raw nMS spectra were visualized using Thermo Xcalibur Qual Browser (version 4.2.47). Data processing and spectra deconvolution were performed using UniDec version 4.2.0[60,61]. The UniDec parameters used were m/z range: 2000–7000; mass range: 25,000–45,000 Da; sample mass every 0.5 Da; smooth charge state distribution, on; peak shape function, Gaussian; and Beta softmax function setting, 20. The expected masses for the de novo synthesized RNA include 94-mer (30,630 Da), 95-mer (30,936 Da), 110-mer (35,776 Da), 114-mer (37,038 Da), and 116-mer (37,671 Da). The mass deviations of the measured masses from the expected masses were within 1 Da or less.

## *Put*EC preparation and cryo-EM grid freezing

Holoenzyme was formed by mixing *Eco* RNAP and *Eco* σ[70] with 1:2 molar ratio and incubating for 15 min at 37 °C, and purified in Superdex 200 Increase 10/300 Increase GL column (Cytiva). Template DNA was amplified in thermocycler. pRAK31 plasmid[62] was used as template DNA for the PCR reaction. The forward and reverse primer sequences (from Macrogen) for the reaction are as follows; Forward primer-5′-GC ATGAATTCCTATTGGTACTTTACATTAA-3′, Reverse primer-5′-CGAAT TGTGAGCGCTCACAATTCTAAAAGCAAAAAAGCCTTC-3′. Holoenzyme

and template DNA were mixed and incubated for 10 min at 37 °C to form RPo. After RPo reconstitution, LacI, which is also purified by size-exclusion chromatography before use, was added and incubated for 10 min for roadblocking. To the mixture, 1 mM rNTP (Cytiva) was added and incubated for 2 min at 37 °C for transcription. The sample was loaded onto zeba spin desalting column (Thermo Fisher) to remove free rNTP, and 2 mM IPTG was added to the complex. After 2 min incubation at 37 °C, the mixture was concentrated using 30 K MWCO Amicon Ultra Centrifugal Filter (Merck Millipore) up to 5–10 μM. The final buffer condition for all cryo-EM samples was 20 mM Tris-glutamate (pH 8.0), 10 mM Mg-glutamate, 150 mM K-glutamate, 5 mM DTT. 0.5% CHAPSO was added to the sample right before grid freezing. For cryo-EM grid freezing, Quantifoil R 1.2/1.3 Cu 400 grids were glow discharged at negative polarity, 0.26 mbar, 15 mA, 25 sec. Using a Vitrobot Mark IV (Thermo Fisher), grids were blotted and plunge-frozen into liquid ethane with 100% chamber humidity at 22 °C.

### Cryo-EM data acquisition and processing
Micrographs were taken using a 300 keV Krios G4 (Thermo Fisher Scientific) with a K3 BioQuantum direct electron detector (Gatan) with 20 eV energy filter slit width. Images were recorded with EPU with a pixel size of 1.06 Å/pix over a defocus range of −0.8 μm to −2.6 μm. Total dose given to the data set is 42.16 $e^-/Å^2$ and total frame number was 55. The movies were drift-corrected, summed, and dose-weighted using MotionCor2 in RELION3.1[63]. The contrast transfer function (CTF) was estimated using Gctf[64], and the summed images were sorted based on CTF max resolution (<10 Å) and CTF figure of merit (>0.01).

The sorted images were transferred to cryoSPARC v3.2.0 for further process[65]. First, 411.9k particles were picked using blob picker from 2000 movies, extracted with 320 pixels box size, and 2D classified to make picking templates. Then, 1447.1k particles were picked using template picker from 8174 images. The particles were 2D classified twice, and the selected 863.1k particles from 43 classes were used as templates for Topaz picker. From Topaz train, 1202.5k particles were picked and extracted from 8162 images. The particles were 2D classified into 100 classes and 90 classes were selected. The selected particles were divided into five classes in heterogeneous refinement. Among the five templates, three are from the previous data set collected from Glacios, two are from EMDB EMD-8585, a non-paused EC map. Among five classes, three classes were subjected to homogeneous refinement. Each homogeneous-refined class was further heterogeneous-refined into two classes, resulting in total of four significant EC classes—RPo, *put*EC, *put*-less EC, and σ70-bound *put*EC.

All particles of the four classes were imported to RELION3.1 for further refinements. The particles belonged to holoenzyme structure were 3D auto-refined, particle-polished three times, and 3D classified into three classes. Among the three classes, the major class was 3D auto-refined and post-processed yielding 3.0 Å-resolution map. The *put*EC particles were 3D auto-refined, particle-polished three times, and subjected to focused classification onto *put*RNA region into three classes. Among the three classes, two classes are combined, 3D auto-refined and post-processed yielding 3.2 Å-resolution map. The *put*-less particles were 3D auto-refined, particle-polished three times, and post-processed yielding 3.6 Å-resolution map. The σ70-bound *put*EC particles were 3D auto-refined, particle-polished three times, and 3D classified into three classes. Among the three classes, one best class was further refined and post-processed yielding 3.6 Å-resolution map.

### Model building, refinement, and validation
The local resolution estimation and filtration were done by blocres and blocfilt commands in bsoft package (version 2.0.5), respectively[66]. For the EC structures, EC coordinates including RNAP, DNA, and RNA are used from PDB 6C6T because this is modeled from the high-resolution EC map (3.5 Å). For σ70-bound *put*EC, the recently published

high-resolution RPo model (PDB 6OUL) was used. In the model building, the models were first fitted onto the final cryo-EM map by using UCSF Chimera (version 1.11.12)[67]. Then, the RNAP domains were rigid-body refined in PHENIX (version 1.18.2)[68], and the nucleic acid were mutated to have the correct sequence in Coot[69]. The structures were then real-space refined in PHENIX, manually modified in Coot, and iterated this process until satisfied. The *put*RNA was manually built into the map de novo. A .eff file that includes restraints maintaining the nucleic acid base pairing and stacking interactions was provided for each real-space refinement run. For the final refinement run, the nonbonded_weight parameter value was set to 500 (default value: 100) to improve the MolProbity and clash scores. The local filtered map was also used for the last refinement iteration because it slightly improved the modeling when inspected by eyes. The figures were made using PyMOL (version 2.4.0).

### Reporting summary
Further information on research design is available in the Nature Research Reporting Summary linked to this article.

## Data availability
The data that support this study are available from the corresponding authors upon reasonable request. Cryo-EM data have been deposited in the Electron Microscopy Data Bank (EMDB) under accession numbers EMD-33466 (*put*EC), EMD-33468 (*put*-less EC), and EMD-33470 (σ70-bound *put*EC). Atomic models have been deposited in the RCSB Protein Data Bank under accession codes 7XUE (*put*EC), 7XUG (*put*-less EC), and 7XUI (σ70-bound *put*EC). The source data underlying Figs. 1c and 3c are provided as a Source Data file. An uncropped scan of gel from Supplementary Fig. 1b is displayed as Supplementary Fig. 15. Source data are provided with this paper.

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

## Acknowledgements

We thank Dr. Jin-Seok Choi at the KAIST Analysis Center for Research Advancement for help with cryo-EM grid screening, Dr. Bum Han Ryu at the Research Solution Center of the Institute of Basic Science (IBS) for help with cryo-EM data collection, Dr. Hanseong Kim at Institute of Membrane Proteins in Pohang for help with preliminary cryo-EM data collection, and Dr. Seth A. Darst and Dr. Elizabeth A. Campbell at the Rockefeller University, and Dr. Max E. Gottesman at the Columbia University for the manuscript reading and helpful advice. Computational works for this research were performed on the data analysis hub, Olaf in the IBS Research Solution Center. This work was supported by the National Research Foundation of Korea, NRF-2019R1F1A1064026, NRF-2019M3E5D6066058, NRF-2021R1C1C100656011 to JYK, and National Institutes of Health P41 GM109824 and P41 GM103314 to BTC. This manuscript is dedicated to the memory of Dr. Robert Weisberg, who laid the intellectual and experimental groundwork for these studies, and whose untimely passing deprived him of the satisfaction of seeing these come to fruition.

## Author contributions

S.H. purified *Eco* RNAP, reconstituted *put*EC, performed radiolabeled transcription assay, collected and analyzed cryo-EM data, and wrote the manuscript. P.D.B.O. performed nMS experiment, analyzed the result, and wrote the manuscript. J.L. helped S.H. in radiolabeled transcription assay, cryo-EM grid preparation, and cryo-EM data collection. J.K. helped S.H. in cryo-EM data analysis. B.T.C. provided nMS facility and funding and reviewed the manuscript. R.A.K provided materials, analyzed the data, and wrote the manuscript. J.Y.K. conceived the project, analyzed the cryo-EM data with S.H., wrote the manuscript with R.A.K., S.H., and P.D.B.O., and obtained funding for the project.

## Competing interests

The authors declare no competing interests.
