## [Peer Review File · Nature Communications]

Structural basis of transcriptional regulation by a nascent RNA element, HK022 putRNAReviewers' Comments:

Reviewer #1:

Remarks to the Author:

The authors report cryo-EM structures of an E. coli transcription elongation complex (TEC) containing the all-RNA transcription antipausing and antitermination factor put (TEC-put) and an E. coli transcription elongation complex containing the transcription initiation factor sigma and put (TEC-sigma-put).

The cryo-EM structures define the secondary structure and tertiary structure of put; define the interactions between put and the TEC (interactions with the RNA polymerase beta' subunit zinc binding domain at the mouth of the RNA polymerase RNA-exit channel); suggest that put inhibits RNA polymerase "swivelling," a small rotation of the beta' zinc binding domain and adjacent modules of RNA polymerase relative to the rest of RNA polymerase that has been hypothesized to be associated with transcriptional pausing; and show that sigma stabilizes put structure and/or interactions.

The results should be of interest to researchers in bacterial transcription and transcriptional regulation. The manuscript should be acceptable for publication after revision to address the following points:

line 76: Replace "Assembly" by "Preparation." (In the transcription field, the term "assembly" connotes reconstitution of a transcription complex from a nucleic-acid scaffold and individual protein components, not preparation of a transcription complex by promoter-dependent transcription initiation.)

line 110: Replace "in vitro assembled putEC" by "putEC prepared by promoter-dependent transcription initiation, transcription elongation, and DNA roadblocking." (See above)

lines 122-12: State whether the presence of put at the mouth of the RNA polymerase RNA-exit channel narrows or otherwise restricts the RNA-exit channel in a manner that would prevent formation of pause and terminator RNA hairpins. Provide a figure and provide specific, concrete details, including the dimensions of the channel and channel mouth without and with put. Cite and describe refs. 43-46, and compare the dimensions of the channel and channel mouth in TEC-put to the dimensions in the structures of the transcription antipausing and antitermination complexes of refs. 43-46.

lines 300-301: Delete sentence. Replace with text citing and describing the sigma-containing TECs of ref. 45, ref. 46, and <https://www.biorxiv.org/content/10.1101/2022.01.24.477500v1>, and comparing the structure and interactions of sigma in TEC-sigma-put to those in the sigma-containing TECs of ref. 45, ref. 46, and <https://www.biorxiv.org/content/10.1101/2022.01.24.477500v1>.

lines 349-420. Expand. Replace the current vague discussion with clear, specific, concrete details, including the dimensions of the channel and channel mouth in TEC -put and in TEC. Compare the dimensions of the channel and channel mouth in TEC-put to the dimensions in the structures of the transcription antipausing and antitermination complexes of refs. 43-46, state a clear and specific inference about whether, and in what ways, the mechanisms of antipausing and antitermination by put are likely to be analogous to, or different from, the mechanisms of the antipausing and antitermination factors of refs. 43-46, and suggest specific experiments to test the inferences.

Reviewer #2:

Remarks to the Author:

Hwang et al. report the first structures of a bacterial transcription complex modified for antitermination by a nascent RNA structure, put, and of a bacterial transcription complex modified by a sigma initiation factor. Both types of transcription complexes are well-described in the field of

bacterial transcriptional regulation. The put-antitermination complex has long been a source of fascination because most antitermination complexes involve protein regulators, but put works solely via a nascent RNA structure. It is likely that the put antitermination complex is paradigm for many other examples of this type of RNA-based regulation that remain to be discovered in diverse bacteria. Sigma modification of elongation complexes is well documented, but its regulatory significance remains poorly understood. Given the high significance of both of these structures, there is little doubt the manuscript merits high-profile publication. The work is expertly conducted and the manuscript is relatively accessible and easy to follow. I have only minor suggestions to offer that the authors may wish to consider in a final revision.

Minor points (listed in order of appearance not significance)

1. line 59 RfaH is introduced without context. It is unlikely the general reader will know much about RfaH. Does ref. 16 even mention RfaH? A brief explanation of how RfaH works and an appropriate citation would be helpful here.
2. line 73. It's a little confusing here to refer to interaction with transcription factors after put is introduced to be factor-independent. A little more context is needed if transcription factors are mentioned.
3. line 231 "swiveling...is a universal feature" of pausing. This is likely an over-statement. There are likely to be paused states that are not swiveled, even though suppression of swiveling may suppress most pauses.
4. line 239 Is the different in SI2 a consequence of put interaction with the flap? If so, that might be noted here.
5. lines 248, 252, 254 and throughout the manuscript. "density map". Technically, cryo-EM maps are not electron density maps and should not be referred to as such.
6. line 300 "...first structure." As a general rule, it's best not to claim a result is first (even when true).
7. line 357 "non-conical" should be "non-canonical"
8. line 368. The analogy to RfaH may be overstated here. The result may be similar (anti-swiveling), but are the put-EC interactions causing anti-swiveling the same as the RfaH-EC interactions causing anti-swiveling? More generally, the authors could improve the manuscript by offering a structural hypothesis for how put prevents swiveling.
9. line 376. The idea that sigma retention causes roadblock readthrough is interesting, but are there possibilities? Is it possible put aids sigma retention?
10. line 381. The suggested mechanism of sigma retention is speculative. It would benefit from some sort of experimental test. Otherwise note that it's speculation. In describing the putative -10 interaction, explicitly state whether the interaction is hypothesized to occur with DNA or with the comparable sequence in RNA. Can the authors exclude a sigma-RNA interaction?
11. line 415 "Alternativley" should be "Alternatively"
12. line 416 – why is it alternatively? Couldn't both mechanisms contribute?
13. Fig. 4. The maps here are reasonably convincing with regard to the active nucleotide model. However, what about the upstream fork junction? The authors claim it includes an 11 bp hybrid. A map of the upstream fork junction region should also be presented to document this claim.

14. Fig. 5. The authors give the amino acids involved in the beta'-CT in the text but they also should be specified in the legend of better yet in the figure. beta'-CT is not a well-known feature. What does CT stand for. Some readers may think it means beta' C-terminal.

Reviewer #3:

Remarks to the Author:

The authors present their structural and biochemical studies regarding putRNA, an RNA element from the HK022 phage that interferes with transcriptional pausing in *E. coli*. The biological relevance of this mechanism relies in the necessity of the virus to bypass the regulations in place in bacterial systems, in which the genes of phages are preceded by transcription termination sites. However, HK022 codes two putRNAs (L and R), which when synthesized by the bacterial RNAP, form complex secondary structures (two stem-loops) that maintain RNAP in its EC conformation and therefore in active transcription. In this study, the authors utilised cryo-EM to solve the structure of putL RNA bound and unbound to RNAP. They found putRNA to interact with the β' ZBD of RNAP and trap it in a non-swiveled conformation which maintains the EC conformation active and therefore hinders pausing from taking place. Additionally, they analyse a population of simultaneously sigma-factor and putL RNA bound RNAP where the sigma factor presumably allows for more time for putL RNA folding to take place.

I wish to state that I am not a bacterial transcription expert but that my expertise is focused on eukaryotic transcription. In that regard, I, however, find the structural characterization of a trapped RNAP bound to a co-transcriptionally-formed and highly structured RNA molecule, very interesting. The study, conducted by Dr Kang and colleagues, could, therefore not only be relevant for the field of bacterial transcription but also for a broader audience in the transcription field and beyond. However, I have some concerns regarding the interpretation and reliability of some of the results that are presented throughout the study, in particular regarding the de-novo building of the putRNA molecule. I think that some strong statements that appear throughout the text cannot be interpreted from the presented results and require further validation and characterization before I could recommend publishing this manuscript.

Major points

- Figure 1c shows a transcription assay to characterize the 7nt scaffold used to stall the RNAP in the absence of putRNA. I initially thought that the authors use this scaffold for a cryo-EM control of put-less EC. However, it appears that the put-less EC structure was obtained via cryo-EM data processing by 3D classification. It is, therefore, confusing to me why the 7nt scaffold was used in the first place. I strongly recommend that the authors clarify the rationale of this experiment further. Also, it would improve clarity if the figure contains the data from the putWT scaffold (also to have a side-by-side comparison with the put- lane shown on the right).

- The putRNA was built de novo. The authors claim that the cryo-EM density is of sufficient quality stating that the estimated local resolution is ca. 3.5 Å. This raises several, potentially problematic, points:

-- First of all, I am not sure if a local resolution of 3.5 Å is sufficient quality to do so. RNA molecules exhibit much more molecular freedom than polypeptides and less restraints can be added. The authors need to provide more close-up views showing the quality of the cryo-EM map-model fits, of both well resolved regions that feature clear secondary structural elements, but also of tricky regions, so that readers can judge the quality of the authors interpretation. Ideally, other papers that used similar strategies and map qualities to de novo build RNA structures need to be cited.

-- Secondly, by inspecting the local resolution estimates in Sup Fig 3c, the local resolution seems to be more in the range of 4.5 to 5.5. I suggest that the authors provide a close-up view of the putRNA region (potentially with a central slice through the density) to back-up their statement that the local

resolution is 3.5 Å in the putRNA region.

- Figure 2c shows a NucPlot of RNA-amino acid interactions. Several salt bridges and H-Bond contacts are depicted. Given that the local resolution cannot be easily judged by inspecting Sup Fig 3c, close-up views of RNA-amino acid interactions need to be shown using the cryo-EM map (in addition to or instead of the calculated surface representation, shown in Figure 3) to show the quality of the map-model fits (with amino acids being shown).

- Sup. Fig 4b shows a comparison between the predicted and the modelled RNA molecule. Given that the modelled RNA clearly differs from the prediction, I suggest that the authors try to place the predicted RNA molecule and compare the fits of the two models. It could well be that the cryo-EM map is simply not of sufficient quality to visualize the predicted hairpin.

- Based on how the cryo-EM data quality is depicted throughout the manuscript, I am not entirely sure if the data is of sufficient quality to confidently build that many non-canonical Watson-Crick base-pairs.

- Given the (in my opinion) uncertainty of the built model, I would tend to suggest that the RNA secondary structure needs to be biochemically validated.

- Figure 3 describes the interaction between putRNA and the β ZBD. I assume only calculated surfaces based on the atomic model are shown. If that is correct, the authors should state this in the figure legends. Getting back to my point raised regarding Figure 2c, I think that the actual cryo-EM densities should be shown in addition or instead.

- Fig. 3 shows a mutational analysis. While the thorough investigation of putRNAs can be appreciated, I don't fully understand how the relative values were derived. The put- mutant shows a pausing activity of 0. When cross-checking with Sup Fig 5a (right panels), it seems that the band intensity for the put- mutant does not correlate well with the relative value of 0. Or was the put- data used to calibrate this value to a relative 0? If so, I can't extract this type of information from the text.

- Based on the absence of error bars in Fig. 3c, I assume the experiments were not repeated multiple times. If yes, I leave it up to the editor to judge if this fits the scientific standards of the journal. Personally, I, however, think the experiments need to be done at least in triplicates and error bars should be shown.

- Figure 4a,b: a conformational change of β SI2 region of the beta-subunit is indicated (between put-RNA and put-less EC structure). However, it appears to me that this could also be due to fragmented density (parts of unmodelled cryo-EM density are visible in this region) of the put-less EC structure. At the chosen threshold level, the loop also doesn't show any density. Neither does the alpha-helix seem to fit the cryo-EM density very well. A better representation of this region (potentially as a close-up view) with depicted cryo-EM densities shown side-by-side might be necessary to convincingly demonstrate the conformational change of this region.

- Table 1. Based on the decent cryo-EM resolution ranges (3.1 to 3.6), I would have expected better MolProbity scores than the given values of 2.22 to 2.41. Also, clash scores are pretty high. Ramachandran statistics are acceptable but I feel they could also be better than 87% to 90% favoured. Hence, I have the impression that the model building, refinement, and validation procedure could have been performed more thoroughly.

Minor points:

- The authors refer to electron density maps, which is not entirely correct for cryo-EM maps. I suggest using a different term throughout the manuscript, such as cryo-EM density.

- Fig. 1c: what does RO mean?
- Fig. 1c is obviously cropped. The sentence, the gel has been cropped for clarity should be added in the figure legends and alongside other figures containing cropped gels.
- Fig. 1d: D is in capital letters and placed behind the image.
- Throughout the text, there is an inconsistency in the font size of figure labels such as "a", "b", "c", etc.
Figure 2a: similar coloration of the RNA and the β' ZBD densities. We suggest considering the choice of more discernible coloration for these two densities.
- Sup Fig. 3a,b require scalebars
- Sup Fig 5 shows a titration (or time course?): 0, 0.5, 2. It should be added in the figure legends or indicated in the figure what this titration (or time course?) means (even if specified in the methods).
- The method section does not specify, which buffers were used during protein purification (except for LacI, which is not the main factor of interest in this study).
- Method section, in vitro transcription: not indicated, how transcription was quenched at the specified time-points.
- Method section model-building/refinement part: refinement strategy should state, which restraints were used during real-space refinement.

List of errata:

Line 277: "have influence on", has

Line 154: "generating 1130.3 A", generating a

Line 39: "discovered in Hongkong in early 1970s", in the early

Line 321: "we only modelled visible part", modelled the visible

Point-to-point response to reviewers

Reviewer #1 (Remarks to the Author):

The authors report cryo-EM structures of an E. coli transcription elongation complex (TEC) containing the all-RNA transcription antipassing and antitermination factor put (TEC-put) and an E. coli transcription elongation complex containing the transcription initiation factor sigma and put (TEC-sigma-put).

The cryo-EM structures define the secondary structure and tertiary structure of put; define the interactions between put and the TEC (interactions with the RNA polymerase beta' subunit zinc binding domain at the mouth of the RNA polymerase RNA-exit channel); suggest that put inhibits RNA polymerase "swiveling," a small rotation of the beta' zinc binding domain and adjacent modules of RNA polymerase relative to the rest of RNA polymerase that has been hypothesized to be associated with transcriptional pausing; and show that sigma stabilizes put structure and/or interactions.

The results should be of interest to researchers in bacterial transcription and transcriptional regulation. The manuscript should be acceptable for publication after revision to address the following points:

→ We first thank the reviewer for writing a considerate summary and providing valuable comments and suggestions in the review. The revised sentences in the manuscript are indicated below and highlighted in red letters in the revised manuscript.

1. line 76: Replace "Assembly" by "Preparation." (In the transcription field, the term "assembly" connotes reconstitution of a transcription complex from a nucleic-acid scaffold and individual protein components, not preparation of a transcription complex by promoter-dependent transcription initiation.)

→ We have changed it (line #77 in the revised manuscript).

2. line 110: Replace "in vitro assembled putEC" by "putEC prepared by promoter-dependent transcription initiation, transcription elongation, and DNA roadblocking." (See above)

→ We have changed it (lines #113-114 in the revised manuscript)

3. lines 122-12: State whether (1) the presence of put at the mouth of the RNA polymerase RNA-exit channel narrows or otherwise restricts the RNA-exit channel in a manner that would prevent formation of pause and terminator RNA hairpins. (2) Provide a figure and provide specific, concrete details, including the dimensions of the channel and channel mouth without and with put. Cite and describe refs. 43-46, and compare the dimensions of the channel and channel mouth in TEC-put to the dimensions in the structures of the transcription antipassing and antitermination complexes of refs. 43-46.

→ (1) The presence of the *put*RNA does not directly block or narrow the RNA exit channel. Instead, the negatively-charged phosphate backbone of *put*RNA is adjacent to the RNA exit channel, presumably hindering the RNA duplex formation in the RNA exit channel. To display this, we modeled a duplex RNA in the RNA exit channel of the *put*EC and found the distance between the phosphate backbones of the modeled RNA hairpin and the *put*RNA is only ~5 Å. To describe this, we have added supplementary Fig. 4 and the sentences below to the manuscript:

In the cryo-EM structure of the putEC, the putRNA was located at the opening of the RNA exit channel of the EC adjacent to the β'ZBD (Fig. 2a). This location is consistent with

the put-inactivating RNAP mutations and would restrict the RNA hairpin formation in the adjoining RNA exit channel via electrostatic repulsion (Supplementary Fig. 4). (lines #123-126 in the revised manuscript).

(2) To provide a more quantitative description of the RNA exit channels of the *putEC* together with other anti-termination complexes, we added Supplementary Fig. 11. In this figure, we defined the mouth of the RNA exit channel by five residues (β 'K395, β K1306, β 'D67, β L845, and β N856), drew a pentagon by connecting the C α s of the residues, measured the area of the pentagon in each anti-termination complex and compare the measurements. Here, the pentagon is not on a perfect 2D plane but flat enough to be used to compare the sizes of the RNA exit channel openings. In the Q21- and P7- anti-termination complexes, Q21 and P7 narrow the mouth. Thus, we picked a residue from each protein that protrudes toward the channel opening and included its C α atom to re-define the area of the channel mouth. The relative estimation of the size of the RNA exit channel entrance showed that the RNA exit channel harboring an RNA hairpin has the largest opening. In contrast, Q21-bound and P7-bound ECs have smaller channel openings, blocking the RNA hairpin formation in the channel. By comparison, *putEC* and λ N-EC have larger RNA exit channel openings than the non-paused EC, implying that these two anti-termination complexes utilize different strategies. We described this in the Discussion section as follows:

*Structural studies on prokaryotic anti-termination complexes including λ N, Q21, Xoo P7, and HK022 put suggest general strategies for anti-termination^{7,46,47,49}. (1) The anti-termination factors inhibit RNA hairpin formation by either narrowing the channel or hindering the RNA hairpin folding (Supplementary Fig. 4, 11). The RNA exit channel is thought to aid RNA hairpin formation by its positively-charged residues located inside the channel³³. In Q21 and Xoo P7 anti-termination complex, the anti-termination factors, Q21 and P7 proteins bind at the mouth of the RNA exit channel and confine the channel (Supplementary Fig. 11). The narrowed RNA exit channel only allows single-stranded RNA to move through it and restricts nascent RNA folding for hairpin-dependent pausing and intrinsic termination. In λ N anti-termination complex, λ N binding to the EC remodels the bound NusA and NusE to destabilize the RNA hairpin folding. In addition, the rearranged Nus factors bind to β flap-tip, which stabilizes RNA hairpin pause, possibly preventing the flap-tip from assisting RNA hairpin pausing and termination⁵³. Like λ N, HK022 put also does not narrow the RNA exit channel directly. Instead, the phosphate backbone of the *putRNA* is located near the RNA exit channel, prohibiting the RNA hairpin formation with its negative-charged surface. Modeling an RNA duplex in the RNA exit channel of *putEC* shows that the phosphate backbones of the modeled RNA duplex and the *putRNA* are just ~5 Å apart from each other (Supplementary Fig. 4). In addition, the β flap-tip binds to the *putRNA*, possibly sequestering it from assisting RNA hairpin pause as in the λ N-antitermination complex. (2) In general, anti-termination proteins stabilize the elongation proficient conformation of EC. λ N transverses the RNAP hybrid cavity stabilizing the active form of the EC and binds to the upstream duplex DNA, enhancing the anti-backtracking and anti-swiveling activity of NusG. In the Q21-EC structure, Q21 binding is not compatible with swiveled conformation. Therefore, Q21 counteracts swiveling, leading to anti-pausing⁴⁹. Our data suggests that *putRNA* also reduces swiveling. This stabilization of the active form of an EC may consolidate the RNA exit channel so that it can no longer accommodate the folding of secondary structures that promote pausing and termination⁵⁴⁻⁵⁶. (lines #423-450 in the revised manuscript)*

4. lines 300-301: Delete sentence. Replace with text citing and describing the sigma-containing TECs of ref. 45, ref. 46, and <https://www.biorxiv.org/content/10.1101/2022.01.24.477500v1>, and comparing the structure and interactions of sigma in TEC-sigma-put to those in the sigma-containing TECs of ref. 45, ref. 46,

and <https://www.biorxiv.org/content/10.1101/2022.01.24.477500v1>.
→ We deleted the sentence and referred to the structures as follows:

Recently, cryo-EM structures of σ^{70} -bound ECs were reported in the context of 21Q- and λ PR²-associated ECs^{46–49}. While these complexes are at the paused state in that the σ_2 domain interacts with a -10-like sequence, our σ^{70} -bound putEC is not in a σ -dependent paused state and contains > 100 base-long RNA having a σ^{70} in a different conformation from those in other σ^{70} -bound ECs (Fig. 1d) (lines #316-321 in the revised manuscript).

5. lines 349-420. Expand. Replace the current vague discussion with clear, specific, concrete details, including the dimensions of the channel and channel mouth in TEC -put ad in TEC. Compare the dimensions of the channel and channel mouth in TEC-put to the dimensions in the structures of the transcription antipassing and antitermination complexes of refs. 43-46, state a clear and specific inferences about whether, and in what ways, the mechanisms of antipassing and antitermination by put are likely to be analogous to, or different from, the mechanisms of the antipassing and antitermination factors of refs. 43-46, and suggest specific experiments to test the inferences.

→ We expanded the discussion based on the reviewer's comments listed above and added supplementary figs. 4 and 11 (lines #423-450 in the revised manuscript). At the end of the Discussion section, we suggested further determining the cryo-EM structure of the putEC at the terminator sequence to demonstrate the effects of put on transcription termination.

Reviewer #2 (Remarks to the Author):

Hwang et al. report the first structures of a bacterial transcription complex modified for antitermination by a nascent RNA structure, put, and of a bacterial transcription complex modified by a sigma initiation factor. Both types of transcription complexes are well-described in the field of bacterial transcriptional regulation. The put-antitermination complex has long been a source of fascination because most antitermination complexes involve protein regulators, but put works solely via a nascent RNA structure. It is likely that the put antitermination complex is paradigm for many other examples of this type of RNA-based regulation that remain to be discovered in diverse bacteria. Sigma modification of elongation complexes is well documented, but its regulatory significance remains poorly understood. Given the high significance of both of these structures, there is little doubt the manuscript merits high-profile publication. The work is expertly conducted and the manuscript is relatively accessible and easy to follow. I have only minor suggestions to offer that the authors may wish to consider in a final revision.

→ We thank the reviewer for the beautiful description of our study and its significance. We also thank the reviewer for the comments and suggestions below. The revised sentences in the manuscript are indicated below and highlighted in red letters in the revised manuscript.

Minor points (listed in order of appearance not significance)

1. line 59 RfaH is introduced without context. It is unlikely the general reader will know much about RfaH. Does ref. 16 even mention RfaH? A brief explanation of how RfaH works and an appropriate citation would be helpful here.

→ We described RfaH in more detail and corrected the reference as follows:

Interestingly, putRNA reduces both backtrack and hairpin-dependent pauses like RfaH¹⁸. RfaH, a paralog of NusG, recognizes an ops (operon polarity suppressor) sequence on the non-template DNA strand loaded onto the RNAP EC, changes its C-terminal helices into a β -

sheet KOW domain fold to become active, and inhibits transcriptional pausing by resisting RNAP swiveling^{19,20} (lines #59-63 in the revised manuscript).

2. line 73. It's a little confusing here to refer to interaction with transcription factors after put is introduced to be factor-independent. A little more context is needed if transcription factors are mentioned.

→ From the cryo-EM data analysis, we learned that most (if not all) σ^{70} -retaining *putEC* passed through the *lacO* sequence blocked by LacI during transcription. This indicates that the presence of σ^{70} in *putEC* enabled the EC to overcome the roadblock protein, LacI, suggesting a possibility that the retained σ^{70} could modulate transcription when there is any protein factor bound on the DNA downstream of the EC. We tried to hint at this by the sentence. To clarify the sentence, we amended it as follows:

*Additionally, the σ^{70} -bound *putEC* structure suggested that σ^{70} binding to EC might facilitate RNA folding as well as play a role in transcription modulation* (lines #72-74 in the revised manuscript).

And we also edited the discussion to describe better that the σ^{70} changed the processivity of the RNAP during elongation as follows:

*Second, the σ^{70} -bound EC was resistant to the LacI roadblock. The σ^{70} -bound *putEC* revealed an extra density for a duplex RNA in the RNA exit channel, suggesting that the retained σ^{70} modified the EC to overcome the roadblock during elongation* (Fig. 1d, 5a, Supplementary Fig. 10b) (lines #418-422 in the revised manuscript).

3. line 231 "swiveling...is a universal feature" of pausing. This is likely an over-statement. There are likely to be paused states that are not swiveled, even though suppression of swiveling may suppress most pauses.

→ We revised the sentence as follows:

*Swiveling was first introduced from the structural study of *hisPEC*, and later revealed in the backtracked EC, implying that the swiveling motion plays an important role in both RNA hairpin pause and backtrack pause*³³⁻³⁵. (lines #246-249 in the revised manuscript).

4. line 239 Is the different in SI2 a consequence of put interaction with the flap? If so, that might be noted here.

→ β SI2 does not have direct interaction with the *putRNA* except β flap-tip-helix. β flap-tip-helix is a mobile motif connected to the flap domain by two flexible linkers. Therefore, we do not think the conformational differences of β SI2 between *putEC* and *put-less EC* result from the interaction between the *putRNA* and the flap. Instead, we believe this reflects the overall state of the RNAP in each complex (active in the *putEC*, paused/inactive in the *put-less EC*) because most known structures of active ECs and paused ECs have distinct conformations of SI2 (less tilted vs. more tilted structures as in the fig. 4b).

5. lines 248, 252, 254 and throughout the manuscript. "density map". Technically, cryo-EM maps are not electron density maps and should not be referred to as such.

→ We changed the term "density map" to "map" throughout the manuscript.

6. line 300 "...first structure." As a general rule, it's best not to claim a result is first (even when true).

→ We agree with the reviewer's comment and deleted the sentence.

7. line 357 “non-conical” should be “non-canonical”

→ We corrected it, and then the sentence is trimmed for the word limits.

8. line 368. The analogy to RfaH may be overstated here. The result may be similar (anti-swiveling), but are the put-EC interactions causing anti-swiveling the same as the RfaH-EC interactions causing anti-swiveling? More generally, the authors could improve the manuscript by offering a structural hypothesis for how put prevents swiveling.

→ We edited the sentence by removing the phrase ‘like RfaH’ as follows:

Together, the structures revealed that putRNA promotes RNA synthesis by resisting swiveling (lines #389-390 in the revised manuscript).

We agree with the reviewer that the structural hypothesis for how *put* prevents swiveling would improve the manuscript. However, the motivation for anti-swiveling (and swiveling) seems complicated. For example, RfaH and NusG counteract swiveling similarly to *put*RNA. However, their binding to EC is entirely different from *put*RNA binding to the EC. We currently can not suggest any hypothesis for the motivation of anti-swiveling action in the manuscript. But we would like to study this in-depth in the future.

9. line 376. The idea that sigma retention causes roadblock readthrough is interesting, but are there possibilities? Is it possible put aids sigma retention?

→ We are also interested in the point. We classified the σ^{70} -bound *put*EC further, and still the RNA duplex in the RNA exit channel, which is evidence of roadblock readthrough, accompanied by the presence of σ^{70} . We do not know how the σ^{70} aids the EC overcome the roadblock. One possible situation is that the bound σ^{70} removed LacI bound on the downstream DNA. Or σ^{70} might increase the processivity of the EC other than the -10-like sequence. We are unsure if this is a specific case for LacI protein or if the σ^{70} -bound *put*EC can remove other proteins bound on the downstream DNA. We would need additional experiments to answer this question.

Regarding the relationship between *put* and σ^{70} retention, we did not find any evidence of physical contact between *put* and σ^{70} from the structure. However, among the complexes imaged by cryo-EM, most σ^{70} -bound EC has well-folded *put* while ~ two-thirds of the σ^{70} -unbound EC has *put*. This implies that the presence of *put* positively correlates with the σ^{70} retention. Then, we would need to ask (1) if *put* promotes σ^{70} retention, and (2) if σ^{70} aids *put* folding. We wrote that σ^{70} binding to the -10-like sequence (+23 to +28 in Fig. 1a) in the *put* sequence could aid *put* folding by providing a time window via σ -dependent pausing in the Discussion section. However, we need additional data to see if the *put* favors the σ retention.

Thanks to the reviewers' comment, we are reminded that we have cryo-EM data of the *put*-EC. The same procedure prepares this sample with *put*EC except that we used *put* DNA scaffold, a validated inactive *put* sequence containing five point mutations in the 43rd-47th positions. The cryo-EM analysis of *put*-EC resulted in an EC map without *put*RNA as expected. Intriguingly, the EC particles were classified into two classes – one with σ^{70} , one without σ^{70} – and σ^{70} -bound EC was ~ 44.6%. In the cryo-EM data analysis of *put*EC, the σ^{70} -bound population among the *put*-containing ECs is ~ 42.0%, which is similar to the σ^{70} -bound EC population in the *put*-EC sample. This suggests that the presence of σ^{70} aids *put* folding while *put* does not aid σ^{70} retention. We added the cryo-EM data analysis workflow in Supplementary fig. 12 and included this discussion in the revised manuscript as follows:

*Furthermore, we examined the ratio between σ^{70} -bound EC and σ^{70} -unbound EC from both the *put*EC sample and the *put*-EC to see if the presence of *put* affects the σ^{70} retention (Supplementary Fig. 12). The *put*-EC was prepared in exactly the same way as the *put*EC*

preparation except the *put* template was used as the DNA scaffold and did not show any well-folded *put*RNA density in the cryo-EM maps. From the cryo-EM data analysis, the percentages of σ^{70} -bound EC in the *put*EC (having intact *put*) and the *put*-EC sample were ~42.0% and ~44.6%, respectively, suggesting that the presence of *put* does not affect σ^{70} retention (lines #411-422 in the revised manuscript).

10. line 381. The suggested mechanism of sigma retention is speculative. It would benefit from some sort of experimental test. Otherwise note that it's speculation. In describing the putative –10 interaction, explicitly state whether the interaction is hypothesized to occur with DNA or with the comparable sequence in RNA. Can the authors exclude a sigma-RNA interaction?

→ As the reviewer pointed out, we speculated the occurrence of σ -dependent pausing from the presence of the -10-like sequence in the *put* gene. We edited the sentences in the discussion section and clearly stated that we suspected the occurrence of σ -dependent pausing from the presence of a -10-like sequence as follows:

*A -10-like sequence is known to induce σ -dependent pausing by engaging its non-template DNA region with the σ_2 domain⁵². We suggest that this sequence may cause σ -dependent pausing which facilitates *put*L folding by providing more time (lines #401-404 in the revised manuscript).*

In addition, the interaction is expected to occur with non-template DNA, as we write in the sentence above. The cryo-EM map of σ^{70} -bound *put*EC did now not reveal any interaction between *put*RNA and σ^{70} . Therefore, we do not suggest any direct interaction between them, although this does not exclude the possibility of the transient interaction between *put* and σ^{70} .

11. line 415 “Alternativley” should be “Alternatively”

→ We corrected. But the corrected sentence was removed while we modified the manuscript following the next comment.

12. line 416 – why is it alternatively? Couldn't both mechanisms contribute?

→ We agree with the reviewer that both mechanisms are not exclusive and can co-exist. We have expanded the paragraph and suggested that those two mechanisms (the narrowing of the RNA exit channel and the consolidation of the active EC conformation) could work together as follows:

*Structural studies on prokaryotic anti-termination complexes including λ N, Q21, Xoo P7, and HK022 put suggest general strategies for anti-termination^{7,46,47,49}. (1) The anti-termination factors inhibit RNA hairpin formation by either narrowing the channel or hindering the RNA hairpin folding (Supplementary Fig. 4, 11). The RNA exit channel is thought to aid RNA hairpin formation by its positively-charged residues located inside the channel³³. In Q21 and Xoo P7 anti-termination complex, the anti-termination factors, Q21 and P7 proteins bind at the mouth of the RNA exit channel and confine the channel (Supplementary Fig. 11). The narrowed RNA exit channel only allows single-stranded RNA to move through it and restricts nascent RNA folding for hairpin-dependent pausing and intrinsic termination. In λ N anti-termination complex, λ N binding to the EC remodels the bound NusA and NusE to destabilize the RNA hairpin folding. In addition, the rearranged Nus factors bind to β flap-tip, which stabilizes RNA hairpin pause, possibly preventing the flap-tip from assisting RNA hairpin pausing and termination⁵³. Like λ N, HK022 put also does not narrow the RNA exit channel directly. Instead, the phosphate backbone of the *put*RNA is located near the RNA exit channel, prohibiting the RNA hairpin formation with its negative-charged surface. Modeling an RNA duplex in the RNA exit channel of *put*EC shows that the phosphate backbones of the modeled RNA duplex and the *put*RNA are*

just ~5 Å apart from each other (Supplementary Fig. 4). In addition, the β flap-tip binds to the putRNA, possibly sequestering it from assisting RNA hairpin pause as in the λ N-antitermination complex. (2) In general, anti-termination proteins stabilize the elongation proficient conformation of EC. λ N transverses the RNAP hybrid cavity stabilizing the active form of the EC and binds to the upstream duplex DNA, enhancing the anti-backtracking and anti-swiveling activity of NusG. In the Q21-EC structure, Q21 binding is not compatible with swiveled conformation. Therefore, Q21 counteracts swiveling, leading to anti-pausing⁴⁹. Our data suggests that putRNA also reduces swiveling. This stabilization of the active form of an EC may consolidate the RNA exit channel so that it can no longer accommodate the folding of secondary structures that promote pausing and termination⁵⁴⁻⁵⁶. (lines #423-450 in the revised manuscript)

13. Fig. 4. The maps here are reasonably convincing with regard to the active nucleotide model. However, what about the upstream fork junction? The authors claim it includes an 11 bp hybrid. A map of the upstream fork junction region should also be presented to document this claim.
→ We added a figure showing an entire RNA-DNA hybrid, lid loop at the upstream-fork junction, and bridge helix to show the 11-bp hybrid in Fig. 4d. The figure label was also added (line #294 in the revised manuscript).

14. Fig. 5. The authors give the amino acids involved in the beta'-CT in the text but they also should be specified in the legend of better yet in the figure. beta'-CT is not a well-known feature. What does CT stand for. Some readers may think it means beta' C-terminal.
→ We agree with the reviewer and changed the term 'beta'-CT' to the original word, beta'-clamp-toe, throughout the manuscript. The domain has been known, but its name was recently coined in the literature, Cartagena et al., 2019, PNAS. In the publication, this beta'-clamp-toe binds to Crl, a positive regulator of σ^S , and this interaction stabilizes the σ^S -RNAP-promoter complex initiation complex promoting transcription initiation. Therefore, beta'-clamp-toe might provide a binding platform for some transcription regulators.

Reviewer #3 (Remarks to the Author):

The authors present their structural and biochemical studies regarding putRNA, an RNA element from the HK022 phage that interferes with transcriptional pausing in *E. coli*. The biological relevance of this mechanism relies in the necessity of the virus to bypass the regulations in place in bacterial systems, in which the genes of phages are preceded by transcription termination sites. However, HK022 codes two putRNAs (L and R), which when synthesized by the bacterial RNAP, form complex secondary structures (two stem-loops) that maintain RNAP in its EC conformation and therefore in active transcription. In this study, the authors utilised cryo-EM to solve the structure of putL RNA bound and unbound to RNAP. They found putRNA to interact with the β' ZBD of RNAP and trap it in a non-swiveled conformation which maintains the EC conformation active and therefore hinders pausing from taking place. Additionally, they analyse a population of simultaneously sigma-factor and putL RNA bound RNAP where the sigma factor presumably allows for more time for putL RNA folding to take place.

I wish to state that I am not a bacterial transcription expert but that my expertise is focused on eukaryotic transcription. In that regard, I, however, find the structural characterization of a trapped RNAP bound to a co-transcriptionally-formed and highly structured RNA molecule, very interesting. The study, conducted by Dr Kang and colleagues, could, therefore not only be relevant for the field of bacterial transcription but also for a broader audience in the transcription field and beyond. However, I have some concerns regarding the interpretation and reliability of

some of the results that are presented throughout the study, in particular regarding the de-novo building of the putRNA molecule. I think that some strong statements that appear throughout the text cannot be interpreted from the presented results and require further validation and characterization before I could recommend publishing this manuscript.

→ We thank the reviewer for the kind evaluation, the helpful comments, and the valuable suggestions on our study. The revised sentences in the manuscript are indicated below and highlighted in red letters in the revised manuscript.

Major points

1. Figure 1c shows a transcription assay to characterize the 7nt scaffold used to stall the RNAP in the absence of putRNA. I initially thought that the authors use this scaffold for a cryo-EM control of put-less EC. However, it appears that the put-less EC structure was obtained via cryo-EM data processing by 3D classification. It is, therefore, confusing to me why the 7nt scaffold was used in the first place. I strongly recommend that the authors clarify the rationale of this experiment further. Also, it would improve clarity if the figure contains the data from the putWT scaffold (also to have a side-by-side comparison with the put- lane shown on the right).

→ We changed the fig. 1c as the reviewer suggested by showing transcription assay lanes for wild-type *put*, 7-nt put, and *put* sequence. And to clearly express the need for the *in vitro* putRNA synthesis method using *Eco* RNAP, we added sentences at the beginning of the first result section as follows:

Because an active form of HK022 putRNA can be produced only by the enzymatic synthesis using host RNAP, we prepared the putRNA-associated EC by initiating RNA synthesis with Eco RNAP holoenzyme and stalling the synthesis using a roadblock protein LacI (lac repressor), as previously described with some optimization for cryo-EM study (Fig. 1)¹⁶ (lines #78-82 in the revised manuscript).

2. The putRNA was built de novo. The authors claim that the cryo-EM density is of sufficient quality stating that the estimated local resolution is ca. 3.5 Å. This raises several, potentially problematic, points:

- First of all, I am not sure if a local resolution of 3.5 Å is sufficient quality to do so. RNA molecules exhibit much more molecular freedom than polypeptides and less restraints can be added. The authors need to provide more close-up views showing the quality of the cryo-EM map-model fits, of both well resolved regions that feature clear secondary structural elements, but also of tricky regions, so that readers can judge the quality of the authors interpretation. Ideally, other papers that used similar strategies and map qualities to de novo build RNA structures need to be cited.

- Secondly, by inspecting the local resolution estimates in Sup Fig 3c, the local resolution seems to be more in the range of 4.5 to 5.5. I suggest that the authors provide a close-up view of the putRNA region (potentially with a central slice through the density) to back-up their statement that the local resolution is 3.5 Å in the putRNA region.

→ We agree with the reviewer that a 3.5 Å-local resolution is insufficient for *de novo* modeling because bases cannot be differentiated under this resolution. However, we have structure prediction and base pairings. To convince readers about our *de novo* modeling, we added supplementary data 2 to describe our *de novo* building procedure. We also added close-up views of the local resolution of the *put* region (Supplementary Fig. 5) and the superimposition of the putRNA density and the modeled coordinates (Supplementary Fig. 6). Although the map on the periphery is ~ 5 Å, the central region and the most complicated region of the map is ~ 3 Å, showing relatively clear traces of the phosphate backbone. Based on the supplementary data 2, we believe our structure is correct for the following reasons:

(1) We could identify the 3'-end of the *put*RNA because there was a connected density between the *put*RNA and the RNA-DNA hybrid in the map. We did not build the model in the connecting region due to the poor density.

(2) We had the prior prediction that there are two stem structures. The second stem (Stem II) location was evident in the map because the map clearly shows a long RNA duplex density at the 3'-end of the *put*RNA. Then the first stem (stem I) was visible, although it was more complicated than we expected due to an unexpected bulge structure in the middle of the stem I and the third strand in the triplex region. In stem I modeling, we located an RNA duplex in stem I. The discontinued phosphate backbone could be connected due to its relatively high resolution ($\sim 3 \text{ \AA}$). We could build the bulge and the triplex structure following the phosphate backbone, filling the *put* density.

(3) The registry based on the predicted structure but modified by cryo-EM density made sense regarding the base pairing.

(4) Our mutagenesis study supports the structure.

2. Figure 2c shows a NucPlot of RNA-amino acid interactions. Several salt bridges and H-bond contacts are depicted. Given that the local resolution cannot be easily judged by inspecting Sup Fig 3c, close-up views of RNA-amino acid interactions need to be shown using the cryo-EM map (in addition to or instead of the calculated surface representation, shown in Figure 3) to show the quality of the map-model fits (with amino acids being shown).

→ In this resolution, we agree that those interactions cannot be clearly defined as salt bridges and hydrogen bonds. However, notifying those potential interactions would help understand the interface between *put*RNA and the RNAP. Therefore, we added a notice in the manuscript as follows:

*Most of the **potential** interactions between the *put*RNA and the RNAP comprise polar interactions such as salt bridges, hydrogen bonds, cation- π interactions, and long-range ionic interactions. Although the resolution of the map is not sufficient to specify these short-range interactions, we suggested possible interactions for reference (Fig. 3b, Supplementary Table 2) (lines #166-170 in the revised manuscript).*

3. Sup. Fig 4b shows a comparison between the predicted and the modelled RNA molecule. Given that the modelled RNA clearly differs from the prediction, I suggest that the authors try to place the predicted RNA molecule and compare the fits of the two models. It could well be that the cryo-EM map is simply not of sufficient quality to visualize the predicted hairpin.

→ We generated an alternative *put*RNA structure following the predicted model in coot, fitting to the cryo-EM density and superimposed it on the *de novo put*RNA model (figure below). As shown in the figure, (1) the predicted *put*RNA model cannot explain the extra density in stem I region, which is the third strand of the RNA triplex in *de novo put*RNA model. In addition, the phosphate backbone trace of the predicted model does not match to the density. (2) In stem II, the middle region has the most discrepancies between the predicted and cryo-EM models. This region forms an interface with RNAP β' ZBD, and the bulge structure in the predicted structure makes clashes with the β' ZBD. Therefore, we believe our *put*RNA model is more reliable than the predicted model regarding the cryo-EM density and the mutagenesis study.

Figure for reviewers

4. Based on how the cryo-EM data quality is depicted throughout the manuscript, I am not entirely sure if the data is of sufficient quality to confidently build that many non-canonical Watson-Crick base-pairs.

→ We added Supplementary data 2 and Supplementary figs. 5, 6, and the figure above to support our *put*RNA structure in addition to our mutagenesis study. We hope these additional materials convince the reviewer and the readers.

5. Given the (in my opinion) uncertainty of the built model, I would tend to suggest that the RNA secondary structure needs to be biochemically validated.

→ We performed a mutational study (Fig. 3c) to convince the structure, and we think the results agree with our structure model to some extent. We sought other biochemical assays to examine the structure, however, could not find it, primarily because of the limitation that the *put*RNA folds only by host RNA polymerase during the transcription from the promoter.

6. Figure 3 describes the interaction between *put*RNA and the β ZBD. I assume only calculated surfaces based on the atomic model are shown. If that is correct, the authors should state this in the figure legends. Getting back to my point raised regarding Figure 2c, I think that the actual cryo-EM densities should be shown in addition or instead.

→ We modified the figure to contain the actual cryo-EM density with the model in figure 3a in the revised manuscript.

7. Fig. 3 shows a mutational analysis. While the thorough investigation of putRNAs can be appreciated, I don't fully understand how the relative values were derived. The put- mutant shows a pausing activity of 0. When cross-checking with Sup Fig 5a (right panels), it seems that the band intensity for the put- mutant does not correlate well with the relative value of 0. Or was the put- data used to calibrate this value to a relative 0? If so, I can't extract this type of information from the text.

→ To clarify the mutational analysis, we added more explanation in the Method section (lines #554-570 in the revised manuscript). As the reviewer can see from the transcription assay gel, *put* does not remove the pausing completely but reduces the pausing compared to the *put* scaffold. Therefore, we initiated transcription with various DNA scaffolds (wild type *put*, *put*⁻, and mutants) by adding rNTP containing ³²P-α-CTP and quenched the reaction at 0, 0.5, and 2 minutes by adding 2x loading dye. To estimate the anti-pausing activity of each scaffold, we first estimated paused transcript fractions at 0.5 min. The paused transcript fraction is calculated by: the paused band intensity/(the paused band intensity + the runoff band intensity)

The band intensities were normalized by the number of cytosines contained in the transcript. Then, we calculated the relative anti-pausing activities of mutants by using this equation:

$$\text{Relative anti-pausing activity of mutant } x = 1 - \frac{(P_x - P_{WT})}{(P_{put^-} - P_{WT})}$$

(P_x = the fraction of the paused band of mutant x)

This equation calculates the relative anti-pausing activities of wild type *put* and *put*⁻ as 1 and 0, respectively. The experiment was done in triplicates, and the standard error was shown in the fig. 3c. We believe this estimation method intuitively reflects the anti-pausing activities of the mutants compared to wild type *put* and the validated inactive mutant *put*⁻.

8. Based on the absence of error bars in Fig. 3c, I assume the experiments were not repeated multiple times. If yes, I leave it up to the editor to judge if this fits the scientific standards of the journal. Personally, I, however, think the experiments need to be done at least in triplicates and error bars should be shown.

→ We performed the experiment in triplicates and added standard error bars. The mean relative anti-pausing activities were slightly changed compared to the previous single experiment, but the overall result is the same, supporting the cryo-EM structure. We modified the paragraph analyzing the result of the mutagenesis study as follows:

Among the twenty-three mutations we generated, eleven mutants showed ≤ 20% anti-pausing activity (named 'inactivating' mutations) and three mutants showed ≥ 90% anti-pausing activity (named 'inert' mutations). The inactivating mutations, Δ3-7*, U28*A, U28*C, G35*A, G35*U, G35*C, G45A*, A64*G, G35*C/A9*G, G35*A/A9*G, and G35*A/A9*U, suggest that (1) the 5'-region (from A3* to G7*) is essential for the anti-pausing activity. A₃GACG₇ and its base-pairing region, U₁₉CUGC₁₅ have relatively high conservation scores of (6,6,4,9,9) and (7,7,5,10,10), respectively. This region is the first RNA duplex formed during the putRNA synthesis, and therefore, may provide a platform for further RNA folding. (2) U28*, which protrudes toward the β'ZBD and binds to a small pocket is essential for the function. Interestingly, while U28*A and U28*C abolish the anti-pausing activity, U28*G retained ~60% of the activity. From the structure, we substituted the U28* with the other bases and found that G can form three hydrogen bonds with the surrounding β' residues while A and C form two and*

one potential hydrogen bonds, corroborating the result of the mutational study (Supplementary Fig. 9b). Interestingly, the original put residue, U28* forms fewer hydrogen bonds than guanine and adenine, but exhibits better activity than these, implying that U28* might have additional role(s) besides binding to the RNAP, or the mutants might have different structures from the modeled ones. (3) All of the G35* mutations we generated abrogated the anti-pausing activity of putRNA. We expected that the double mutants, G35*C/A9*G, and G35*A/A9*G might have some activity because they preserve the predicted base-pairing of G35*-A9* in the structure. However, mutating G35* to any base abrogated the anti-pausing activity and this was not recovered by the mutation of the base-pairing partner, implying that G35*, and possibly its base-pairing partner A9*, may have sequence-specific roles in the anti-pausing activity. We noticed that G35*U exhibited ~70% anti-termination activity *in vivo*¹². This discrepancy could come from the different conditions encountered *in vivo* vs. *in vitro*. For example, the G35*U might form some intact or partially active putRNA *in vivo*, possibly aided by an unknown cellular factor(s) whereas *in vitro* synthesized putRNA containing G35U* could be inactive. (4) We also found that A64* is critical for the anti-pausing activity. This result is also consistent with the structural data because it contacts the stem I region of putRNA and the RNAP. All the inert mutations are of U20*, which lacks any significant interaction with other residues, supporting our structure. The remaining nine mutants exhibited moderate activities suggesting a significant, but not critical role of the residues (A8*, U21*, C25*, U32*, G43*). In summary, our mutagenesis study supports our cryo-EM structure of the putEC (lines #192-226 in the revised manuscript).

9. Figure 4a,b: a conformational change of β SI2 region of the beta-subunit is indicated (between put-RNA and put-less EC structure). However, it appears to me that this could also be due to fragmented density (parts of unmodelled cryo-EM density are visible in this region) of the put-less EC structure. At the chosen threshold level, the loop also doesn't show any density. Neither does the alpha-helix seem to fit the cryo-EM density very well. A better representation of this region (potentially as a close-up view) with depicted cryo-EM densities shown side-by-side might be necessary to convincingly demonstrate the conformational change of this region. → As the reviewer pointed out, put-less EC contains a small density behind β SI2, looking like an alternative conformation of β SI2. However, the figure below shows that the modeled β SI2 structure is more dominant than the other possible β SI2 conformation. The density in the terminal loop region is relatively weak, but the density for the helices reveals the prominent location of the β SI2. Thus, we think the observation and speculation are still valid. We have generated the figure the reviewer suggested and attached it below.

putEC**put-less EC**
2. Table 1. Based on the decent cryo-EM resolution ranges (3.1 to 3.6), I would have expected better MolProbity scores than the given values of 2.22 to 2.41. Also, clash scores are pretty high. Ramachandran statistics are acceptable but I feel they could also be better than 87% to 90% favoured. Hence, I have the impression that the model building, refinement, and validation procedure could have been performed more thoroughly.

→ To improve the validation scores, we modified Ramachandran outliers and clashing atoms and increased the nonbonded weight parameter in the Phenix real-space refinement. This improved the scores a lot. We added the improved parameters and scores in Table 1, and added here for reference.

	putEC (PDB 7XUE/EMD-33466)	put-less EC (PDB 7XUG/EMD-33468)	σ^{70}-bound putEC (PDB 7XUI/EMD-33470)
Validation			
MolProbity score	1.69	1.89	2.07
Clashscore	5.27	6.66	8.99
Favored rotamers (%)	99.89	99.37	99.63
Poor rotamers (%)	0	0.11	0
Ramachandran plot			
Favored (%)	93.84	91.02	88.29
Allowed (%)	6.16	8.92	11.68
Disallowed (%)	0	0.06	0.03

Minor points:

1. The authors refer to electron density maps, which is not entirely correct for cryo-EM maps. I suggest using a different term throughout the manuscript, such as cryo-EM density.

→ We changed the term throughout the manuscript.

2. Fig. 1c: what does RO mean?

→ RO stands for run-off. We added this abbreviation in the figure legend (line #721).

3. Fig. 1c is obviously cropped. The sentence, the gel has been cropped for clarity should be added in the figure legends and alongside other figures containing cropped gels.

→ We performed the transcription assay again to make uncropped gel (Fig. 1c). In addition, we examined the activity of the roadblocked EC by adding IPTG and additional rNTP. We did this assay to check if our cryo-EM sample would be in an active condition, but forgot to put it on our manuscript. Thanks to the reviewer's comment, we had added this lane in the Fig.1c. We modified the Method section (lines #712-721 in the revised manuscript) and the figure legend (lines #700-709 in the revised manuscript) accordingly.

4. Fig. 1d: D is in capital letters and placed behind the image.

→ We corrected it

5. Throughout the text, there is an inconsistency in the font size of figure labels such as "a", "b", "c", etc.

→ We corrected it

6. Figure 2a: similar coloration of the RNA and the β' ZBD densities. We suggest considering the choice of more discernible coloration for these two densities.

7. Sup Fig. 3a,b require scalebars

→ We added the scale bars.

8. Sup Fig 5 shows a titration (or time course?): 0, 0.5, 2. It should be added in the figure legends or indicated in the figure what this titration (or time course?) means (even if specified in the methods).

→ We explained the time points in the figure legend.

9. The method section does not specify, which buffers were used during protein purification (except for Lacl, which is not the main factor of interest in this study).

→ We added the conditions.

10. Method section, in vitro transcription: not indicated, how transcription was quenched at the specified time-points.

→ We described more in details.

11. Method section model-building/refinement part: refinement strategy should state, which restrains were used during real-space refinement.

→ We added more details on the refinement.

List of errata:

Line 277: "have influence on", has

Line 154: "generating 1130.3 Å", generating a

Line 39: "discovered in Hongkong in early 1970s", in the early

Line 321: "we only modelled visible part", modelled the visible

→ We corrected these. We again thank the reviewer for the comments.

Reviewers' Comments:

Reviewer #1:

Remarks to the Author:

The revised manuscript has been substantially improved.

It should be acceptable for publication after very minor revision as follows:

1) The structure of a λ antitermination complex was published during the period between review and revision of this manuscript (NusA-containing λ engaged complex of preprint at <https://www.biorxiv.org/content/10.1101/2022.03.25.485794v1>). This structure should be cited and included in Fig. S11, in the discussion of Fig. 11, and in the sentence "Recently, cryo-EM structures of σ 70-bound ECs were reported in the context of 21Q- and λ PR'-associated ECs."

2) Replace "would restrict" by "potentially would restrict" in the newly added sentence "This location is consistent with the put-inactivating RNAP mutations and would restrict the RNA hairpin formation in the adjoining RNA exit channel via electrostatic repulsion (Supplementary Fig. 4)."

3) Replace "plays" by "potentially plays" in the newly added sentence "Swiveling was first introduced from the structural study of hisPEC, and later revealed in the backtracked EC, implying that the swiveling motion plays an important role in both RNA hairpin pause and backtrack pause."

Reviewer #2:

Remarks to the Author:

The authors' revisions to the manuscript adequately address all concerns raised by reviewers. The revised manuscript is suitable for publication.

Reviewer #3:

Remarks to the Author:

Remarks on the revised version of the manuscript by Hwang et al.

I would like to thank the authors for their detailed responses and for having thoroughly addressed the points that I raised during the peer review process.

I find the strategy of putRNA modeling building, together with the newly prepared supplementary figures, convincing and, now, well-described. Hence, the authors have addressed my main concern. I also acknowledge that the in vitro transcription experiments were repeated in triplicates and that cryo-EM densities, instead of calculated surface representations, are shown in Figure 3.

There are only a couple of points (mostly minor suggestions) that I have. Apart from that, I recommend that the manuscript can be published.

Major point:

-In my previous remarks, I asked the authors to include more details regarding the Refinement procedure. Although the authors mention in their rebuttal that they have added more details to the methods section, I cannot find this additional information. I, therefore, ask the authors again to add more details regarding the Refinement procedure. I find it important to state if any restraints were used for real-space refinement in Phenix; in particular for the modeled putRNA.

Minor points:

- Line 129: I suggest referring to Suppl Data 2 for details on the de novo model building.
- Line 169: I suggest referring not only to Figure 3b but also to Figure 2c regarding "possible interaction points"
- Personally, I find the figure, which was prepared for the reviewers and shows the comparison between the modeled and the initially predicted putRNA informative. I, therefore, suggest including this Figure in the supplement and also referring to it in the main text.

Reviewer #4:
None

Point-to-Point response

Reviewer #1 (Remarks to the Author):

The revised manuscript has been substantially improved.

It should be acceptable for publication after very minor revision as follows:

→ We first thank the reviewer for spending time reviewing our manuscript and giving valuable comments and suggestions that improved our manuscript.

1) The structure of a Qlambda antitermination complex was published during the period between review and revision of this manuscript (NusA-containing Qlambda engaged complex of preprint at <https://www.biorxiv.org/content/10.1101/2022.03.25.485794v1>). This structure should be cited and included in Fig. S11, in the discussion of Fig. 11, and in the sentence "Recently, cryo-EM structures of σ^{70} -bound ECs were reported in the context of 21Q- and λ PR'-associated ECs."

→ We added the reference to the sentence as follows:

Recently, cryo-EM structures of σ^{70} -bound ECs were reported in the context of 21Q-, λ PR', and Q λ -associated ECs⁴⁶⁻⁵⁰. (lines #301-302 in the revised manuscript)

In addition, we noted NusA-containing Q λ -engaged EC structure in the discussion as follows:

Structural studies on prokaryotic anti-termination complexes including λ N, Q21, Xoo P7, Q λ and HK022 put suggest general strategies for anti-termination^{7,46,47,49,50}. (lines #401-402 in the revised manuscript)

The structure of Q λ -engaged pTEC is not available yet because they are held for release from the PDB. Therefore, we could not update the supplementary fig. 13 (the figure number is changed from 11 to 13 during the revision). We added this to the figure legend, as well.

2) Replace "would restrict" by "potentially would restrict" in the newly added sentence "This location is consistent with the put-inactivating RNAP mutations and would restrict the RNA hairpin formation in the adjoining RNA exit channel via electrostatic repulsion (Supplementary Fig. 4)."

→ We added 'potentially' to the sentence (line #121 in the revised manuscript).

3) Replace "plays" by "potentially plays" in the newly added sentence "Swiveling was first introduced from the structural study of hisPEC, and later revealed in the backtracked EC, implying that the swiveling motion plays an important role in both RNA hairpin pause and backtrack pause."

→ We added 'potentially' to the sentence (line #237 in the revised manuscript).

Reviewer #2 (Remarks to the Author):

The authors' revisions to the manuscript adequately address all concerns raised by reviewers. The revised manuscript is suitable for publication.

→ We again thank the reviewer for spending time reviewing our manuscript and giving valuable comments and suggestions that improved our manuscript.

Reviewer #3 (Remarks to the Author):

Remarks on the revised version of the manuscript by Hwang et al.

I would like to thank the authors for their detailed responses and for having thoroughly addressed the points that I raised during the peer review process. I find the strategy of putRNA modeling building, together with the newly prepared supplementary figures, convincing and, now, well-described. Hence, the authors have addressed my main concern. I also acknowledge that the in vitro transcription experiments were repeated in triplicates and that cryo-EM densities, instead of calculated surface representations, are shown in Figure 3. There are only a couple of points (mostly minor suggestions) that I have. Apart from that, I recommend that the manuscript can be published.

→ We again thank the reviewer for the valuable comments and suggestions that improved our manuscript.

Major point:

-In my previous remarks, I asked the authors to include more details regarding the Refinement procedure. Although the authors mention in their rebuttal that they have added more details to the methods section, I cannot find this additional information. I, therefore, ask the authors again to add more details regarding the Refinement procedure. I find it important to state if any restraints were used for real-space refinement in Phenix; in particular for the modeled *putRNA*.

→ We apologize for the mistake. We added more details to the method section as follows (line #630-635 in the revised manuscript).

A .eff file that includes restraints maintaining the nucleic acid base pairing and stacking interactions was provided for each real-space refinement run. For the final refinement run, the nonbonded_weight parameter value was set to 500 (default value: 100) to improve the molprobity and clash scores. The local filtered map was also used for the last refinement iteration because it slightly improved the modeling when inspected by eyes.

Minor points:

- Line 129: I suggest referring to Suppl Data 2 for details on the de novo model building.

→ We added it (line #125 in the revised manuscript)

- Line 169: I suggest referring not only to Figure 3b but also to Figure 2c regarding "possible interaction points"

→ We added it (line #163 in the revised manuscript)

- Personally, I find the figure, which was prepared for the reviewers and shows the comparison between the modeled and the initially predicted putRNA informative. I, therefore, suggest including this Figure in the supplement and also referring to it in the main text.

→ We added the figure to the supplementary information as a supplementary fig. 7 and referred to in the main text (line #125 in the revised version). The later figures were renumbered accordingly.